# MMPD-Bench: Bridging Multimodal Fission with Multi-Polarimetric Modalities Decomposition

Yi He [*1]  Zimo Zhao [*2]  Yiming Yang [1]  Xiaoyuan Cheng [1]  Chao He [+2]  Yukun Hu [+1]

## Abstract

Recovering multiple physical parameters from high-dimensional optical measurements remains challenging in computational optics. We present *MMPD-Bench*, a pioneering benchmark that reframes multi-polarimetric modalities decomposition from Mueller matrix observations as a *modality fission* problem under the multimodal learning paradigm. By replacing iterative numerical inversion with deep surrogate models, MMPD-Bench provides data, standardized solutions and evaluations to address the multi-physics modalities generation challenge. We benchmark representative architectures to this problem, including state space models, vision transformers, conditional diffusion models, and neural operators, under a multi-faceted evaluation protocol that jointly assesses perceptual fidelity, physical consistency, robustness, and computational efficiency. Our analysis reveals non-trivial tradeoffs between accuracy and robustness in accelerated high-fidelity polarimetric decomposition, highlighting key limitations of existing surrogates. We open-source a large-scale dataset of 21,412 high-resolution Mueller matrix observations and 4 specialized test sets acquired in physical polarimetric measurements (with code in `https://mmpdbench.github.io`), inviting further research at the intersection of polarization optics and multimodal representation learning.

## 1. Introduction

Scattered light encodes a wealth of information about microstructural organization, composition, and optical properties. To capture this information, Mueller polarimetry measures the complete transformation of the polarization state based on the theoretical Stokes-Mueller formalism. Extracting physical insights from these measurements requires Mueller Matrix Polar Decomposition (MMPD) (Lu & Chipman, 1996; He et al., 2021). Conventionally due to computational-expensive inverse algorithms to obtain polarimetric parameters, MMPD is treated as a rigid task of scientific computing given the expanding resolution of optical observations (Steelman et al., 2019; Li et al., 2023a; Moriconi et al., 2024; Chae et al., 2025). However, viewed through the lens of modern representation learning, we reframe this decomposition as *Modality Fission* in multimodal learning (Vidal, 2011; Abavisani & Patel, 2018; Aktukmak et al., 2019; Chen et al., 2021; Liang et al., 2024). Unlike fusion, which aggregates sensors into a single representation, fission requires the disentanglement of a single entangled observation (the $4 \times 4$ Mueller matrix) into multiple, physically distinct manifolds, including but not limited to diattenuation, retardance, and depolarization. While recent works (Si et al., 2022; Yang et al., 2022; Moriconi et al., 2024; Zhang et al., 2025b; Zheng et al., 2025) have approached this as a generative challenge, generic computer vision datasets fall short in conveying the unique high-dimensional Stokes-Mueller physics (He et al., 2019; Zhang et al., 2025a). Meanwhile, the lack of standardized evaluation on the physical fidelity, robustness, and consistency across different architectures necessitates a dedicated benchmark tailored for such polarimetric inverse problems.

Despite the key drivers above, two distinct requirements that remain challenging for standard architectures on the MMPD task. First, to preserve global spatial continuity of observation objectives, the model should capture long-range dependencies that span the entire field of view rather than focusing on local textures (Nguyen et al., 2024; Ma et al., 2024; Xing et al., 2025). Second, it should capture inter-channel physical dependencies, ensuring that modalities generated remain coupled by the laws of polarization rather than behaving as independent color channels (Li et al.,

---

[1]Dynamic Systems Lab, University College London, London, UK [2]Vectorial Optics and Photonics Group, University of Oxford, Oxford, UK. [*]Yi He <yi.he.20@ucl.ac.uk> and [*]Zimo Zhao <zimo.zhao@eng.ox.ac.uk> contributed equally to this work. [+]Correspondence to: Chao He <chao.he@eng.ox.ac.uk>, Yukun Hu <yukun.hu@ucl.ac.uk>.

*Proceedings of the 43rd International Conference on Machine Learning*, Seoul, South Korea. PMLR 306, 2026. Copyright 2026 by the author(s).

2023a; Moriconi et al., 2024; Yang et al., 2024; Chae et al., 2025). To address these challenges, three primary methods have been reviewed in multi-physics modalities generation from complexed scientific observations: (1) **Deterministic Vision Models**, which excel in learning direct mappings from Mueller matrices to decomposed modalities (LeCun & Bengio, 1998; Li et al., 2017; Zhao et al., 2020; Ma et al., 2021; Yang et al., 2024; Tian et al., 2025; Chen et al., 2025; Dosovitskiy et al., 2021; Liu et al., 2021b;a; 2024a; Zhu et al., 2024; Liu et al., 2024b; Wei & Zhang, 2023; He et al., 2025b; Li et al., 2023b; Xing et al., 2025); (2) **Deep Probabilistic Generative Models**, which model the complex distributions of polarimetric modalities to enable high-fidelity generation through conditioning and iterative denoising processes (Ho et al., 2020; Song et al., 2020a; Si et al., 2022; Choi et al., 2022; Yang et al., 2023; Rissanen et al., 2022; Moriconi et al., 2024; Zhang et al., 2025b; Zheng et al., 2025); and (3) **Neural Operators**, though they have not been introduced to MMPD problems in the literature, these methods demonstrating a broad success in learning differential equations in physical systems (Li et al., 2020; Kovachki et al., 2023; Azizzadenesheli et al., 2024). By learning mappings between function spaces, neural opeator models offer a promising avenue for capturing the continuous nature of polarimetric transformations. More of the motivations and research gaps from these related works are presented in Appendix C.

In this study, we introduce MMPD-Bench, a comprehensive benchmarking framework schematically illustrated in Figure 1. Beyond the foundational dataset contribution, we highlight the following aspects:

1. **Unifying MMPD under ML context**: we formally define MMPD as a *modality fission* problem, establishing a novel bridge that connects high-dimensional Mueller matrix decomposition into a manageable and standardized multimodal generation task in section 2 and 3.

2. **Introducing Neural Operators to MMPD**: we pioneer the adaptation of Neural Operator methods for MMPD to learn mappings between function spaces, building a more comprehensive MMPD-Bench that spans attention, state-space, generative, and operator learning paradigms in section 2.3.4 and 4.

3. **Identifying eight crucial findings**: we provide the first extensive comparative analysis of diverse modern architectures, ranging from classic and linear-complexity attention models to conditional diffusion models. As one of the key insights, we identify a distinct performance stratification where standard vision surrogates and Neural Operators offer superior speed and precision for quantitative metrology, while generative models excel in robustness under perturbations, revealing

critical future outlooks for methodology innovation and development.

4. **Multi-faceted evaluation standard**: we propose a rigorous evaluation protocol that transcends standard vision metrics by incorporating physical consistency checks, numeric matching, and statistical distances to measure modality alignment. This is released as an open-source platform to accelerate the adoption of trustworthy neural surrogates in computational optics, and to provide unique data sources for multimodal fission research in the community.

## 2. Problem Settings

This section lays the theoretical groundwork for MMPD-Bench. We first frame the problem of MMPD as both a *physics inverse problem* in polarization studies and a *multimodal fission* task in representative learning. We categorize the selected methods reviewed in section 1 and Appendix C for benchmarking as **P**ola**r**imetric **V**ision **D**ecomposition (PiVD) methods. Then, we define the adaptation for MMPD using PiVD backbones.

### 2.1. Mueller Matrix and its Polar Decomposition

Governed by the Stokes-Mueller formalism elaborated in Appendix B, we define the MMPD in Lu & Chipman (1996) as a physics inverse problem:

**Definition 2.1** (**MMPD Inverse Formulation**). Let $\mathbf{M} \in \mathbb{R}^{4\times4}$ be the observed Mueller matrix obtained from imaging polarimetry. We define the *forward map* $\mathcal{F}$ that maps the optical parameters $\mathbf{p}$ in Lu & Chipman (1996) to the observed matrix space:

$$\mathcal{F} : \mathbf{p} = \{D, \Delta, \eta, \theta, \psi, R\} \rightarrow \mathbf{M}. \tag{1}$$

Under the inverse problem framework, the objective of MMPD is to solve $\hat{\mathbf{p}} = \mathcal{F}^{-1}(\mathbf{M})$, disentangling the coupled optical effects to recover distinct physical modalities. Solving this inverse problem provides high-contrast diagnostic information invisible to standard intensity imaging, effectively decoding the micro-optical signature of the sample.

### 2.2. Modality Fission

Contextualizing MMPD within the broader machine learning framework, we invoke the concept of Representation *Fusion* and **Fission** from multimodal learning.

**Modality Fission as the inverse of Fusion.** Let $\mathcal{U} = \{\mathbf{u}_1, \ldots, \mathbf{u}_K\}$ represent a set of distinct, disentangled modalities in the subspaces, where each $\mathbf{u}_k \in \mathcal{X}_k$. We define **Fusion** as a forward composition map $\Phi$:

$$\Phi : \mathcal{X}_1 \times \mathcal{X}_2 \times \cdots \times \mathcal{X}_K \rightarrow \mathcal{Z}, \quad \mathbf{z} = \Phi(\mathbf{u}_1, \ldots, \mathbf{u}_K),$$

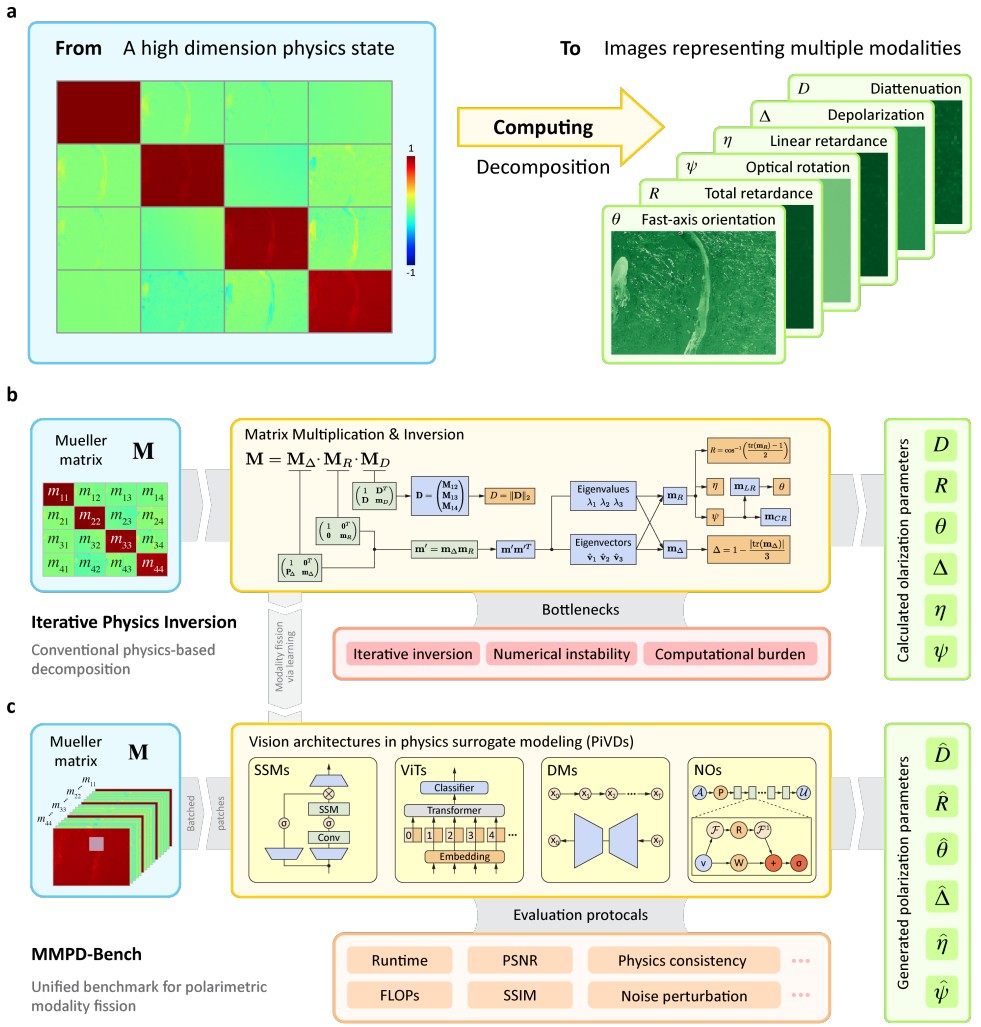

*Figure 1.* **Overview of MMPD-Bench.** (a) A spatially resolved $4 \times 4$ Mueller matrix observation describes the transformation of the polarisation state under the Stokes–Mueller formalism and is decomposed into physically interpretable parameters, including diattenuation ($D$), depolarisation ($\Delta$), linear and total retardance ($\eta$ and $R$), fast-axis orientation ($\theta$), and optical rotation ($\psi$). (b) Conventional MMPD factorises the measured Mueller matrix and conducts physics-based numerical inversion to derive the components, but this process can introduce numerical instability and computational burden for large-scale polarimetric imaging. (c) MMPD-Bench reframes this process as a modality-fission problem and benchmarks deep surrogate models, including state space models (SSMs), vision transformers (ViTs), diffusion models (DMs), and neural operators (NOs), using unified evaluations of fidelity, statistical alignment, physical consistency, robustness, and efficiency.

where $\mathbf{z} \in \mathcal{Z}$ is the observed complex data representation. Conversely, **Fission** is defined as the inverse mapping $\Psi \approx \Phi^{-1}$, aiming to decompose the unified representation $\mathbf{z}$ back into its disjoint components (Liang et al., 2024):

$$\Psi : \mathcal{Z} \to \mathcal{X}_1 \times \cdots \times \mathcal{X}_K, \quad \text{s.t.} \quad \Psi(\mathbf{z}) = \{\hat{\mathbf{u}}_1, \ldots, \hat{\mathbf{u}}_K\} \approx \mathcal{U}.$$

**MMPD as physical Modality Fission.** In the context of Mueller polarimetry, the observed Mueller matrix $\mathbf{M}$ acts as the fused representation in space $\mathcal{Z}$. The constituent polarimetric modalities $\mathbf{p}$ correspond to the disentangled subspaces $\mathcal{X}_k$. The Fusion process $\Phi$ is physically defined in Lu & Chipman (1996) formalism that we elaborate in Appendix B. It couples the modalities via non-commutative

matrix multiplication:

$$\Phi(\mathbf{p}) = \mathbf{M}_\Delta \cdot \mathbf{M}_R \cdot \mathbf{M}_D = \mathbf{M}. \quad (2)$$

The goal is to learn the fission process $\Psi_\theta$ that disentangles the input $\mathbf{M}$ into its constituent polarimetric subspaces:

$$\Psi_\theta(\mathbf{M}) \to \{\hat{D}, \hat{\Delta}, \hat{\eta}, \hat{\theta}, \hat{\psi}, \hat{R}\}. \quad (3)$$

### 2.3. Polarimetric Vision Decomposition (PiVD)

We frame MMPD as a sequence modeling problem to capture long-range spatial correlations within a MM observation. Let the visual representation be denoted as $\mathbf{x}(i) \in \mathbb{R}^d$, where $i \in \{1, \ldots, L\}$ indexes the spatial sequential tokens

derived from the patchified Mueller matrix. The step $i$ corresponds to the spatial location, which assumes the global polarimetric context as a continuous trajectory.

### 2.3.1. STATE SPACE MODELS FOR MMPD

State Space Models (SSMs) learn the interaction between the polarimetric input sequence and a latent memory state. We focus on the discrete formulation regarding the spatial index $i$ to align with the spatial domain token embedding of the Mueller matrix as

$$\mathbf{h}_{i+1} = \bar{\mathbf{A}}_i\mathbf{h}_i + \bar{\mathbf{B}}_i\mathbf{x}_i, \qquad \mathbf{p}_i = \mathbf{C}_i\mathbf{h}_i + \mathbf{D}\mathbf{x}_i, \quad (4)$$

where $\mathbf{h}_i \in \mathbb{R}^n$ is the latent state capturing the accumulated spatial context, and $\mathbf{p}_i \in \mathbb{R}^m$ is the output projection corresponding to the decomposed physical parameters. The matrices $\mathbf{A} \in \mathbb{R}^{n \times n}$, $\mathbf{B} \in \mathbb{R}^{n \times d}$, and $\mathbf{D} \in \mathbb{R}^{m \times d}$ are learnable parameters, while $\mathbf{C}_i$ is input-dependent under the selective-scan mechanism. $\bar{\mathbf{A}}_i$ and $\bar{\mathbf{B}}_i$ are discretized from $\mathbf{A}$ and $\mathbf{B}$ specified in the Vision Mamba integration.

**Vision Mamba integration.** To handle the non-causal nature of optical scattering in images, we adopt the VMamba architecture by Liu et al. (2024b). In VMamba, the SSM is obtained by discretizing an underlying continuous-time system with an input-dependent step size $\Delta_i$, defining $\bar{\mathbf{A}}_i = \exp(\Delta_i\mathbf{A})$ and $\bar{\mathbf{B}}_i = \left(\int_0^{\Delta_i} \exp(\tau\mathbf{A})\,d\tau\right)\mathbf{B}$. This approach allows the model to dynamically filter information based on local observation turbidity. Furthermore, the sequence is processed bi-directionally (Liu et al., 2024b; Zhu et al., 2024), ensuring that the decomposition $\mathbf{p}(i)$ at any location integrates information from the entire global structure.

### 2.3.2. VISION TRANSFORMERS FOR MMPD

Unlike the recurrent state evolution in Eq. 4 which compresses context into a hidden state $\mathbf{h}(i)$, ViTs model dependencies via a global interaction mechanism with Scaled Dot-Product Attention. It computes pairwise similarity between all tokens simultaneously:

$$\text{Attention}(\mathbf{Q}, \mathbf{K}, \mathbf{V}) = \text{softmax}\left(\frac{\mathbf{Q}\mathbf{K}^\top}{\sqrt{d_k}}\right)\mathbf{V}, \quad (5)$$

where queries $\mathbf{Q}$, keys $\mathbf{K}$, and values $\mathbf{V}$ are linear projections of the input sequence $\mathbf{X}$. Unlike linear SSMs, this method builds an $L \times L$ map to capture global structures, resulting in quadratic $\mathcal{O}(L^2)$ complexity.

**Patch-based and Efficient-attention ViTs.** To address computational costs, we adopt two strategies: (1) Patch-based ViTs (Dosovitskiy et al., 2021; Liu et al., 2021a; 2024a), which utilize the full attention mechanism for maximum fidelity in each patch; and (2)

Efficient-attention ViTs (Wang et al., 2020; Choromanski et al., 2020; Li et al., 2021; 2023b), which approximate the self-attention matrix using linear or low-rank projections to achieve scalable complexity.

### 2.3.3. DIFFUSION MODELS FOR MMPD

We treat the decomposition of MM as a *conditional generative* task. We employ Score-based Diffusion Models (SDMs) to approximate the conditional distribution of the physical modalities $\mathbf{p}$ given the fused observation $\mathbf{M}$:

$$p_\theta(\mathbf{p} \mid \mathbf{M}) \approx p_{\text{data}}(\Psi_\theta(\mathbf{M})) \quad (6)$$

**Conditional reverse process.** Leveraging the continuous-time formulation (Song et al., 2020b; Ho et al., 2020), we bypass the standard forward diffusion description and focus on the generative reverse-time SDE. The model iteratively refines a noise sample $\mathbf{x}_T \sim \mathcal{N}(\mathbf{0}, \mathbf{I})$ into the target modalities $\mathbf{p}$ conditioned on $\mathbf{M}$:

$$d\mathbf{x}_t = \left[f(\mathbf{x}_t, t) - g(t)^2\nabla_\mathbf{x} \log p_t(\mathbf{x}_t \mid \mathbf{M})\right]dt + g(t)d\bar{\mathbf{w}}_t \quad (7)$$

where $\nabla_\mathbf{x} \log p_t(\mathbf{x}_t \mid \mathbf{M})$ is the conditional score function approximated by a neural network, allowing the integrated $\mathbf{M}$ to guide the fission subspaces tied to optical phenomena.

**Physical fidelity via classifier-Free guidance.** We employ Classifier-Free Guidance (Ho & Salimans, 2022) to explicitly control the trade-off between generative diversity and physical fidelity. During sampling, the score estimate is modified as:

$$\hat{\epsilon}_\theta(\mathbf{x}_t, \mathbf{M}) = (1 + w)\epsilon_\theta(\mathbf{x}_t, \mathbf{M}) - w\epsilon_\theta(\mathbf{x}_t, \emptyset) \quad (8)$$

where $w$ is the guidance scale that controls the adherence to the conditioning $\mathbf{M}$. This balances the generated sample's alignment with specific conditioning, while maintaining the stochastic generation conformity in the Stokes-Mueller polarization context.

### 2.3.4. NEURAL OPERATORS FOR MMPD

NOs treat the observed Mueller matrix $\mathbf{M}$ and the target physical modalities $\mathbf{p}$ as continuous fields defined on a bounded spatial domain $\Omega \subset \mathbb{R}^2$, mapping from $\Omega$ to matrix manifold $M \in \mathbb{R}^{4 \times 4}$ and manifold $\mathbf{p} \in \mathbb{R}^6$. We formulate the fission task as learning an operator $\mathcal{G}_\theta$ that maps the input function space $\mathcal{A}$ (observation fields) to the output function space $\mathcal{U}$ (physical parameter fields).

**Continuous fission mapping.** Let $a \in \mathcal{A}$ represent the continuous MM field $\mathbf{M}(x)$ and $u \in \mathcal{U}$ represent the decomposed properties $\mathbf{p}(x)$ for any spatial coordinate $x \in \Omega$. The neural operator approximates the inverse fission mapping $\Psi$

defined in Eq 3:

$$\mathcal{G}_\theta : \mathcal{A} \to \mathcal{U}, \quad \text{s.t.} \quad \mathcal{G}_\theta(a)(x) \approx u(x) = \Psi(a(x)). \quad (9)$$

The operator $\mathcal{G}_\theta$ is also resolution agnostic and can be evaluated on arbitrary discretizations of the domain $\Omega$. The Fourier Neural Operator (FNO) (Li et al., 2020) is composed of a sequence of iterative spectral convolution layers indexed by $i$. For a hidden representation $v_i(x)$ at layer $i$, the update rule is defined as:

$$v_{i+1}(x) = \sigma\left(W v_i(x) + (\mathcal{K} v_i)(x)\right), \quad (10)$$

where $\sigma$ is a non-linear activation, $W$ is a local linear transformation, and $\mathcal{K}$ is the convolution operator defined in the spectral domain. By applying the Fast Fourier Transform ($\mathcal{F}$), the global integration becomes a multiplication by a learnable weight tensor $R_i$ in frequency space:

$$(\mathcal{K} v_i)(x) = \mathcal{F}^{-1}\left(R_i \cdot \mathcal{F}(v_i)\right)(x). \quad (11)$$

This parameterization allows the model to learn global spatial correlations by efficiently acting on frequency modes.

## 3. Benchmarks

In this section, we introduce the formulation of MMPD-Bench. We first describe the **datasets** in section 3.1 and Appendix A, including the MM acquisition methods, specific testsets, and polarimetric pairs preprocessing steps. Then, we formulate three **challenges** of polarization decomposition problems: solver surrogate efficiency, multi-modality fidelity and consistency, and robustness under perturbation in section 3.2.

### 3.1. Data

Acquired through labor-intensive, multi-shot Mueller polarimetric measurements, the real-world dataset captures non-ideal effects, modality coupling, and measurement noise that are intrinsic to practical systems but difficult to reproduce synthetically. The samples span a broad range of structural organisations, from ordered birefringent features in healthy tissue to increased depolarisation associated with pathological disruption, providing a challenging and representative testbed for benchmarking Mueller matrix polar decomposition under real-world conditions. A summary of the main bone cell dataset metadata, including resolution, patch configuration, and split statistics, is provided in Table 1[1]. Two specific test data details are provided in Appendix A.

---

[1] Patch counts do not follow a simple $(1 - \text{overlap ratio})$ rule. It refers to the acquisition stage translated with one-directional overlap. Only consecutive captures may partially overlap, and overlapping regions are removed before patch extraction.

**Observation specimens.** The raw data consists of full-field MM observations obtained from prepared tissue sections of healthy and diseased bone (train&test), and standard waveplates (test). All main samples were prepared following standard tissue sectioning protocols (approx. 6 $\mu m$ thickness) and mounted on microscopy slides to ensure stable optical conditions.

*Table 1.* Polarimetric bone cell datasets in MMPD-Bench.

| Attribute | Healthy | Diseased |
|---|---|---|
| Raw Source Slides | 60 | 36 |
| Raw Resolution | $3000 \times 4000$ | $3000 \times 4000$ |
| Overlap Ratio | 40% | 33% |
| *Preprocessing* | | |
| Patch Resolution | $256 \times 256$ | $256 \times 256$ |
| Clip Stride | 200 | 200 |
| Intensity Threshold | $8/256$ | $5/256$ |
| *Data Splits (Patches)* | | |
| Train | 11,294 | 6,006 |
| Validation | 1,453 | 713 |
| Test | 1,303 | 643 |
| **Total Patches** | **14,050** | **7,362** |

**Data acquisition.** The large-scale polarimetric data were acquired using a custom-built wide-field Mueller matrix polarimeter operating in a transmissive optical layout. The main biological dataset was measured at 617 nm using a red LED source, liquid crystal variable retarders (LCVR) to modulate polarization states, and a monochrome CMOS sensor to capture high-resolution field-of-views (3000 $\times$ 4000 pixels). In addition to the primary biological measurements, reference waveplate samples and multi-wavelength measurements were acquired as specialised external test data to evaluate cross-sample and spectral generalisation. Detailed hardware specifications, calibration procedures, measured samples, and acquisition protocols are provided in Appendix A. The corresponding reference polarimetric parameter maps were obtained using the Lu-Chipman decomposition procedure described in Appendix B.

**Preprocessing and partitioning.** To prepare the data for deep learning tasks, the high-resolution raw observations were patchified into $256 \times 256 \times 16$ tensor blocks. We applied intensity-based filtering to exclude low-signal patches. We employed a overlap section clip and a sliding window approach with a defined rate in Table 1 to maximize data yield while preserving local structural context. This process resulted in a total of 21,412 paired samples. The final dataset was stratified with about 80% training, 10% validation, and 10% testing splits, ensuring balanced partitions.

## 3.2. Tasks → Evaluation

Our benchmark is organized around three core tasks, each corresponding to a distinct evaluation metric that designed to assess the utility of PiVD models from computational, physical, and robust perspectives. For each task, specialized evaluation method and metrics are presented to quantify performance. Detailed metric definitions and calculation procedures are provided in Appendix F.

### 3.2.1. COMPUTATIONAL SURROGATE EFFICIENCY

*Objective:* The primary motivation for PiVD is to replace iterative numerical solvers with rapid inference models. This task evaluates the extent to which a model can reproduce the output of a reference physics-based solver while providing significant computational acceleration.

*Evaluation:* We assess this trade-off using two categories of metrics regarding **computational scalability** and **surrogate fidelity**. Firstly, we include (1) the **inference runtime** (seconds/batch) across increasing batch sizes (e.g., BS=1 to 184) to evaluate latency scaling behavior, and We report the acceleration in bar plots, comparing the model's inference time against the baseline numerical Physics Solver; (2) we also include **FLOPs per forward pass** and **peak memory usage** of model inference to determine the theoretical computational cost and GPU memory footprint critical for edge constraints in science labs. Furthermore, to ensure efficiency does not compromise physical correctness, we verify that the surrogate model matches the reference solver outputs using standard reconstruction metrics, specifically **peak signal-to-noise ratio (PSNR)** for pixel-level accuracy and **structural similarity index (SSIM)** for evaluating the preservation of structural information in the physical modalities.

### 3.2.2. MODALITY FIDELITY AND CROSS-MODALITY LEVEL PHYSICS CONSISTENCY

*Objective:* The PiVD baselines are targeted to generate accurate and qualified physical parameters from the observed MM. This task focuses on evaluating the fidelity of the reconstructed modalities with respect to the reference physical decomposition, and assesses whether cross-modality dependencies imposed by polarimetric physics are preserved.

*Evaluation:* We assess the quality of the generated modalities $\hat{p}$ against the reference numerical solver decompositions $p$ through a unified framework combining **vision-based** and **physics-based** metrics. To quantify reconstruction fidelity, we re-utilize the PSNR and SSIM for pixel-wise and perceptual agreement. Meanwhile, we employ the **1D Wasserstein distance** (WD-1d) to ensure the generated distributions align with the ground truth. Beyond visual reconstruction, we evaluate physical consistency by measuring

the cross-modality alignment error derived from Stokes-Mueller properties, specifically focusing on the **retardance consistency** with its derived parameters to verify adherence to the underlying optical laws.

### 3.2.3. ROBUSTNESS TO ACQUISITION PERTURBATIONS

*Objective:* Real-world polarimetry measurements experience sensor noise and calibration drifts. This task investigates how model performance degrades under increasing acquisition noise, in comparison to the reference physics-based solver.

*Evaluation:* We stress-test models by injecting additive Gaussian noise into the input to simulate potential sensor noise, calibration uncertainty and blurs during data acquisition. We train and evaluate the models on the noised observation. We report the above fidelity metrics and further derive the **relative performance degradation rate**, presenting the variance of PiVD methods in fidelity and statistical matching decays. We choose the noise with the standard deviation set to 10% of the pixel value standard deviation ($\sigma_{\text{noise}} = 0.1\sigma_{\text{pixel}}$), which is sufficient to challenge feature extraction methods while maintaining the structural integrity in most acquisition scenarios.

*Table 2.* Overview of benchmarked PiVD architectures.

| Class | Model | Key notes |
|---|---|---|
| SSMs | SwinUMamba (Liu et al., 2024a) | Swin encoder with Mamba blocks for efficient dense prediction. |
| | SegMamba (Xing et al., 2025) | Mamba based model designed for long range modeling with linear scaling. |
| ViTs | SwinUnet (Cao et al., 2022) | Shifted window attention encoder with a U-Net decoder. |
| | FactFormer (Li et al., 2023b) | Factorized attention for scalable global region at lower cost. |
| DMs | DDPM (Ho et al., 2020) | Denoising diffusion with stochastic reverse sampling. |
| | DDIM (Song et al., 2020a) | Deterministic sampling for faster inference. |
| NOs | FNO (Li et al., 2020) | Fourier domain convolution for operator learning. |
| | UNO (Rahman et al., 2023) | U-shaped FNO for multiscale operator learning. |

## 4. Experiments

In this section, we address the research questions outlined in section 3.2 by empirically training and evaluating the performance of various PiVD models on the benchmark datasets. Each model is trained and evaluated with 3 random seeds, given training details in Appendix D. Diffusion models are fully denoised for 1000 sample steps in inference. We present an overview of model performance in Figure 2, followed by a detailed analysis of the results and findings. Details for implementation and decomposition snapshots can be found in Appendices E and G.

### 4.1. Models

We evaluate representative architectures from four categories: (1) State Space Models (SSMs), (2) Vision Transformers (ViTs), (3) Diffusion Models (DMs), and (4) Neural Operators (NOs). For each category, we select two representative architectures; as listed in Table 2. To evaluate scaling effects, each architecture is implemented in two configurations: Small (-s) with reduced parameters and Base (-b) with standard capacity.

### 4.2. Findings

We summarize 8 key findings regarding the objectives in section 3.2 and present critical insights at the intersection of MMPD surrogates and multi-modality generation.

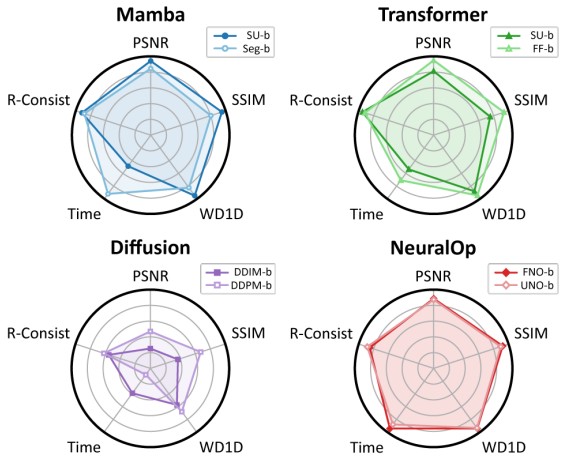

*Figure 2.* Multi-dimensional performance analysis across modalities for models (-b) against 5 metrics on clear observations - 1. PSNR ↑: Pixel-wise reconstruction fidelity and signal quality; 2. SSIM ↑: Preservation of structural information and spatial patterns; 3. WD1D ↓: Global statistical alignment across the entire test population; 4.Time ↓: runtime to inference per sample (for diffusion models, this plot compares the runtime to denoise one step); 5. R-Consist ↑: consistency to the physical retardance.

**Efficiency of surrogates.** We report the multi-dimensional analysis of runtime scalability in Figure 6; and performance trade-offs in Figures 2 and 7. We highlight two critical findings regarding the feasibility of these architectures as real-time physics surrogates:

*(1) Swin-Unet transformers offer superior batch scalability, while Mamba and FactFormer suffer from Out-Of-Memory (OOM) constraints.* As shown in the runtime analysis in Figure 6 measured on the same device, NOs achieve the highest inference efficiency in single-sample processing. Transformer architectures with memory-efficient windowed attention mechanisms demonstrate strong scalability as batch sizes increase. Conversely, *linear-complexity architectures* like Mamba and FactFormer suffer Out-Of-Memory

(OOM) constraints significantly earlier than standard Transformers. Referring to the computational costs in Table 6, Mamba and FactFormer are theoretically efficient in terms of FLOPs, but their high-throughput inference relies on parallel scan algorithms. This results in a sharp increase in peak memory usage compared to Swin-Unet transformers.

*(2) Tailored architectures for MMPD benefit more than scaling model size.* We find that domain-specific designs achieve a superior balance between runtime and accuracy. As illustrated in the radar plots (Figure 2 and 7), compact and specialized architectures, such as FNO-s and UNO-s for learning in the spectral domain, achieve outstanding efficiency in MMPD surrogate modeling. In contrast, scaling to larger variants incurs latency penalties from both OOM constraints and the amplification of architecture complexities. The representative example is the diffusion models: while their backbone networks may be computationally efficient for a single denoising step, the intrinsic **sequential sampling scheme** necessitates multi-step iteration to ensure high-fidelity generation. Consequently, the cumulative inference latency becomes orders of magnitude higher than single-pass deterministic surrogates, rendering them less suitable for real-time physics computing.

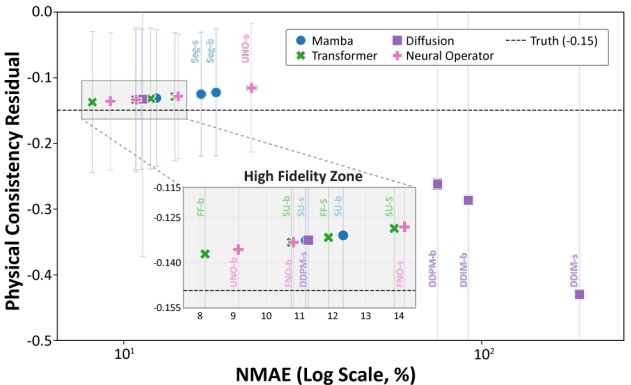

*Figure 3.* Retardance consistency evaluates the physical validity of the generated decompositions by correlating the normalized mean absolute error against the physical consistency residual derived from the retardance (R, $\eta$) and orientation ($\psi$) relationship. The High Fidelity Zone insert highlights models that achieve both high visual accuracy and strict adherence to Stokes-Mueller physics.

**Fidelity and cross-modality consistency.** From the quantitative benchmarks in Table 8 and 11, physical alignment analysis in Figure 3, we identify three critical findings in the trade-offs between deterministic and generative surrogates:

*(1) Deterministic methods outweighs generative priors in precise scientific modalities metrology.* Factformer, FNO, and SwinU Mamba consistently dominate fidelity and numerical evaluation of physical consistency benchmarks, demonstrating strict adherence to Stokes-Mueller physical principles. Diffusion models, in contrast, show the limita-

tions of quantitative physical modality generation. Looking into the generated samples e.g. in Figure 8, the noise residuals explain the exact pixel-level mismatch in the generation.

***(2) FactFormer and Neural Operators resolve angular phase learning challenge.*** A distinct challenge in polarimetric fission is recovering the angular fields ($\theta$ and $\psi$), which contain intrinsic phase discontinuities. From Table 8 and 9, FactFormer and Neural Operators significantly outperform other baselines, with **UNO-b** achieving the highest PSNR for ImageTheta (27.3 dB) and that of **FF-b** for ImagePsi (52.5 dB). This confirms the spectral parameterization in these architectures is more capable in preserving phase structures and the global physical topology than patch-based attention mechanisms and SSMs.

***(3) Diffusion models struggle to recover multi-polarimetric modalities in MMPD.*** Although diffusion models excel at perceptual image restoration, their performance on the fission modalities in MMPD is significantly hindered. We analyzed this phenomenon and retrieved to four fundamental factors. The first factor roots in the *nonlinear error amplification*: MMPD involves highly non-linear operations to derive modalities as presented in Appendix B). Diffusion models as the MMPD surrogates, small noise residuals can cause exponential amplification in the deep layers approximating inversion calculations. Secondly, *the hallucination violates the intrinsic physics*: without specified physics constraints on model training and inference, diffusion models prioritize learning the visual distribution of the data rather than the strict physical laws governing it. As seen in the Retardance Consistency analysis in Figure 3, diffusion models fall far outside the High Fidelity Zone, exhibiting notable consistency residuals across modalities $\{R, \eta, \psi\}$. Thirdly, *incoherent inter-channel denoising*: standard diffusion processes often fail to maintain the algebraic ratios between the 16 channels showing in Figure 5. Given this channel-wise discrepancy, isolating each pure optical property precisely is challenging. Finally, *data sparsity in high-dimensional manifold*: unlike RGB images, MMs exist on a complex, high-dimensional physical manifold. Although 21,412 high-resolution MMs are provided, it is still in the limited size that hinders the model from fully learning the polarization geometry. Researching on data efficient diffusion methods to generalize on sparse scientific observation is crucial for MMPD and the multi-modality fission problems.

**Robustness.** Table 3 reveals three key findings regarding model robustness under acquisition perturbations:

***(1) Increased model scale does not imply improved robustness.*** Smaller model variants (-s) consistently exhibit lower degradation rates in PSNR and SSIM than larger models (-b), despite their lower fidelity performance with clear observations. This indicates that larger models are more

sensitive to high-frequency noise present in Mueller matrix measurements, leading to inferior robustness under noisy conditions.

***(2) U-shaped models are vulnerable to noise.*** Mamba and neural operator models with U-shaped designs show notable performance degradation of up to 49.8% in PSNR, 63.7% in SSIM, and 83.2% in WD, reflecting the trade-off of skip connections that preserve high-frequency information. Transformer-based architectures, by contrast, maintain substantially more stable performance. Due to global attention mechanisms that suppress local noise while preserving modality-level structure, results are consistent to Figure 9.

***(3) Diffusion models perform better with noisy data.*** Diffusion-based surrogates show improved performance with existing measurement noise, as evidenced by increased PSNR and SSIM compared to the noiseless task. Figure 5 reveals that noiseless MM data exhibits a highly peaked, sharp distribution, which can destabilize the score estimation in diffusion models. Adding measurement noise provides a smoothing effect, making the data more consistent with the stochastic perturbations used in diffusion training and generation (Ho et al., 2020; Song et al., 2020b). However, this gain is not general: angular modalities such as ImageTheta remain noise sensitive, which is attributed to the nonlinear arctan in equation 27 behaving sharp changes for a transform that amplifies small residual errors.

## 5. Conclusion

In this work, we introduced MMPD-Bench, the first unified benchmarking with $21,412$ real-world MM observation-decomposition pairs and a multifaceted evaluation standard for deep learning surrogates for MMPD. By defining tasks across computational efficiency, physical fidelity, and robustness, we have provided a holistic methodology to evaluate the readiness of AI models for optical physics applications. Our results reveal that the primary bottleneck for deploying AI surrogates in MMPD is not the computational throughput, but the preservation of physical admissibility and parameter consistency across multiple decomposition outputs. We further observe that different polarimetric parameters and PiVD models exhibit markedly different learnability and robustness under AI approximation, calling for cross-modality alignment beyond single-modality error metrics. In addition, the findings also inspire future diffusion model research works in accelerating denoising processes while optimizing for sparse observations and data efficient learning.

**Limitations and future work.** MMPD-Bench is built primarily on real-world biological Mueller matrix measurements, which reflect practical polarimetric imaging conditions. Reference modalities used in this benchmark are solver-derived outputs obtained from Lu & Chipman (1996)

*Table 3.* The relative degradation (%) across PSNR, SSIM, WD-1d metrics is measured from Table 8, 9, 10 and 11. This table compares how **robust** the models behave under clear observation and with Gaussian noised ($\sigma = 0.05$) observation. Positive values indicate performance degradation (the minimal degradation rates are highlighted, and the maximal ones are underlined). Negative values indicating improvement even under noise. The distinct behavior of generative models and the high sensitivity of ImageTheta is investigated in section 4.2.

| Models | ImageD | | | ImageTheta | | | ImageR | | | ImageDelta | | |
|---|---|---|---|---|---|---|---|---|---|---|---|---|
| | PSNR | SSIM | WD | PSNR | SSIM | WD | PSNR | SSIM | WD | PSNR | SSIM | WD |
| Mamba (SU-s) | 6.3 | 3.5 | 58.6 | 26.1 | 52.6 | 50.3 | 0.2 | 1.0 | 6.2 | 5.4 | 0.6 | 5.6 |
| Mamba (SU-b) | 11.5 | 4.8 | 50.0 | 43.1 | 57.1 | 50.0 | 1.9 | 1.0 | 12.7 | 7.8 | 0.7 | 60.0 |
| Mamba (Seg-s) | 6.7 | 3.4 | 5.1 | 13.5 | 48.8 | 40.4 | 3.4 | 3.0 | 35.4 | 11.4 | 0.6 | 6.5 |
| Mamba (Seg-b) | 7.6 | 6.3 | 49.3 | 14.7 | 49.3 | 46.2 | 1.3 | 0.9 | 13.7 | 11.5 | 1.4 | 38.1 |
| Transformer (SU-s) | 1.0 | 0.4 | 7.9 | 1.4 | 13.9 | 0.0 | 1.1 | 1.1 | 5.0 | 3.6 | 1.3 | 21.1 |
| Transformer (SU-b) | 2.8 | 1.8 | 34.0 | 6.6 | 32.5 | 57.1 | 1.1 | 1.7 | 14.8 | 3.2 | 0.6 | 58.8 |
| Transformer (FF-s) | 0.7 | 0.4 | 11.1 | 39.5 | 56.2 | 60.3 | 1.9 | 0.1 | 8.6 | 6.0 | 0.8 | 46.7 |
| Transformer (FF-b) | 8.4 | 2.4 | 10.0 | 40.7 | 56.4 | 77.5 | 3.1 | 3.9 | 31.5 | 11.8 | 1.1 | 25.0 |
| Diffusion (DDIM-s) | -82.1 | -61.1 | -86.2 | 23.6 | 58.1 | -52.4 | -66.5 | -25.8 | -82.9 | -72.4 | -43.6 | -83.8 |
| Diffusion (DDIM-b) | -45.2 | -39.6 | -85.4 | 43.1 | 65.9 | 39.2 | -45.3 | -20.5 | -74.1 | -50.2 | -4.7 | -80.5 |
| Diffusion (DDPM-s) | -5.3 | 2.3 | -18.8 | 34.4 | 63.3 | 58.5 | -18.5 | -11.8 | -62.4 | 36.8 | 24.8 | 6.6 |
| Diffusion (DDPM-b) | -21.5 | 1.1 | -69.1 | 46.0 | 68.8 | 26.8 | -14.0 | 22.0 | -54.4 | -21.3 | 15.1 | -49.3 |
| NeuralOp (FNO-s) | 1.2 | 2.1 | 19.4 | 29.4 | 56.3 | 47.7 | 2.4 | 1.8 | 4.9 | 1.4 | 1.1 | 23.4 |
| NeuralOp (FNO-b) | 0.2 | 0.2 | 6.8 | 44.8 | 57.9 | 81.4 | 2.5 | 1.8 | 8.8 | 4.5 | 0.8 | 16.7 |
| NeuralOp (UNO-s) | 0.5 | 0.6 | 11.9 | 46.5 | 66.6 | 78.7 | 3.7 | 2.1 | 1.6 | 1.4 | 0.1 | 5.6 |
| NeuralOp (UNO-b) | 0.2 | 1.1 | 2.1 | 49.8 | 63.7 | 83.2 | 3.6 | 3.4 | 21.6 | 9.2 | 2.0 | 30.0 |

framework; future extensions with calibrated phantoms and independently characterised materials would enable more validation of accuracy. Beyond validation, the benefits of MMPD-Bench will be amplified in adaptive optics systems, where fast and reliable polarimetric parameter extraction can support system calibration, performance assessment, and vectorial aberration correction (Zhao et al., 2025; Ma et al., 2025). PiVD models could futher provide the measurement and computational decoding needed for reconfigurable retarder arrays (He et al., 2025a), and future photonic-computing platforms based on spatially varying polarisation fields, where robust information processing requires reliable optical-state readout (Wang et al., 2024; Zhang et al., 2026; Liu et al., 2026). In addition, integrating physics guidance, physical constraints, and problem-specific priors (Sun et al., 2025; Cheng et al., 2025b) into PiVD backbones remains important for improving reliability and consistency; while text annotations and richer metadata may support broader multimodal representation learning, including the use of vision-language models for more diversified interpretation. Together, these extensions would strengthen MMPD-Bench as a reusable framework for polarimetric, representation learning and engineering problems.

## Impact Statement

This paper presents a benchmark for multi-modality decomposition in polarimetric imaging. The datasets used in this study consist of de-identified measurements of biological tissue. MMPD-Bench supports practical deployment in settings where polarimetric imaging is informative but computationally demanding, such as large-area biomedical imaging, high-throughput medical inspection, material characterisation, and non-destructive measurements (He et al., 2021; Moriconi et al., 2024; Chae et al., 2025). More broadly, the modality-fission formulation connects to inverse problems in which a single observation is decomposed into multiple interacting latent factors. This perspective is relevant to representation learning and world-model construction, where disentangling physically or semantically meaningful components can improve interpretability and downstream reasoning. All data samples were rigorously anonymized prior to model training and analysis, removing metadata related to personally identifiable information. The proposed methods are designed strictly for scientific metrology and multimodal representation, with no intended use in surveillance or data profiling.

## Acknowledgment

Yi He acknowledges support of Dean's Prize and Chadwick Scholarship from the University College London. Chao He acknowledges support from the University of Oxford and the Royal Society (URF/R1/241734). Yukun Hu acknowledges support from UCL Innovation Pipeline Funding. The authors thank Prof. Daniel Royston from the University of Oxford and Prof. Honghui He from Tsinghua University for their support to this work.

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

## A. Benchmark Dataset Details

This section provides complementary information about the benchmark datasets used in this study, including the data acquisition system, measured samples, calibration procedure, and pre-processing steps used for the defined tasks. In addition to the primary biological dataset, we describe the reference waveplate measurements and multi-wavelength measurements used as specialised external test data for evaluating cross-sample and -wavelength generalisation.

### A.1. Data Acquisition

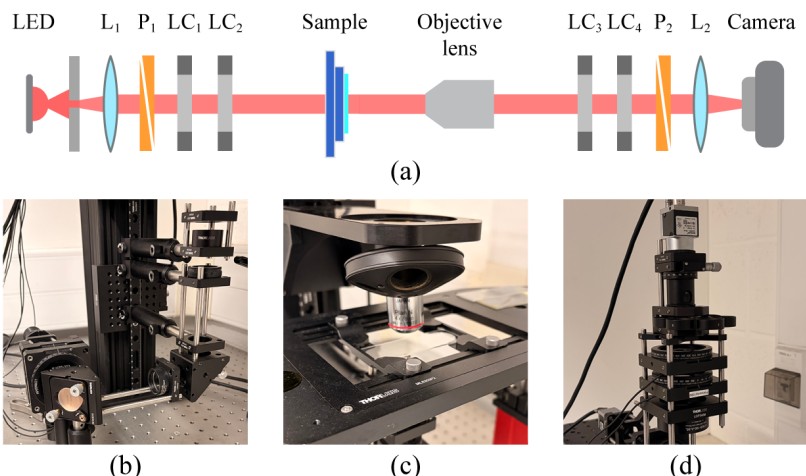

(a)

(b)                    (c)                    (d)

*Figure 4.* Wide-field Mueller matrix polarimetric imaging system used for data acquisition. (a) Schematic of the transmissive optical layout, illustrating the sequential propagation of illumination through the polarisation state generator (PSG), sample, objective lens, and polarisation state analyser (PSA) before detection. (b) Implementation of the PSG, consisting of a linear polariser ($P_1$) followed by two liquid-crystal retarders ($LC_1$, $LC_2$), used to generate a set of predefined input polarisation states. (c) The sample imaging region, illustrating the transmissive wide-field measurement geometry over which spatially resolved polarimetric data were acquired. (d) Implementation of the PSA, composed of two liquid-crystal retarders ($LC_3$, $LC_4$) and an analyser polariser ($P_2$), used to project the output polarisation states prior to camera.

Polarimetric data for this study were acquired using a wide-field Mueller matrix polarimeter (Figure 4) comprising a red LED source (M617L3, Thorlabs), Glan–Taylor polarisers (GT10-A, Thorlabs), and two pairs of uncompensated full-wave liquid-crystal retarders (LCC1223-A, Thorlabs) configured as a polarisation state generator (PSG) and a polarisation state analyser (PSA). The system was operated in a transmissive optical layout, where collimated illumination at a central wavelength of 617 nm sequentially passed through the PSG, the sample, the objective lens (Olympus 4×/0.1 NA), and the PSA before being recorded by a monochrome CMOS camera (acA4024-29um, Basler), producing an effective spatial sampling of $3000 \times 4000$ pixels at the sample plane (Ma et al., 2025; Zhang et al., 2025a). System calibration was performed using air as the reference (Cai et al., 2025), which ideally corresponds to an identity Mueller matrix. A reference dataset was acquired, and the deviation from identity was quantified using the Frobenius-norm metric $E = \|M_{\text{measured}} - I\|_F$ (Lu & Chipman, 1996; Azzam, 2016), with the resulting correction applied to normalise subsequent measurements. Multi-shot averaging, consisting of 5 shots per state with an exposure time of 3 ms, was applied to reduce measurement noise and improve channel consistency. In addition to camera exposure, each polarimetric state transition required the liquid-crystal retarders to switch and settle to their target retardance, resulting in a total acquisition time of approximately 3 s per field of view.

The main biological dataset was acquired from prepared bone-cell tissue sections, including healthy and diseased specimens. The tissue sections were prepared following standard tissue sectioning protocols (Gömöri, 1937), with an approximate thickness of 6 $\mu$m, and mounted on standard microscopy slides (25 mm $\times$ 75 mm $\times$ 1 mm) to ensure stable transmission measurements. Under the calibrated acquisition configuration, the system operated in a multi-shot mode to generate 16-channel Mueller matrix images for diseased bone cells ($12 \times 3$ field-of-views) and healthy bone cells ($15 \times 4$ field-of-views). In addition to the biological specimens, reference waveplate samples, including commercial half- and quarter-wave plates (WPH10M-633 and WPQ10M-633, Thorlabs), were measured as controlled reference measurements with retardance-dominant polarimetric responses. The biological measurements form the primary train/validation/test dataset of MMPD-Bench, whereas the waveplate measurements are used as specialised external test data for evaluating cross-sample

generalisation.

To acquire multi-wavelength measurements, the red LED illumination source was replaced by a white-light source (MN-WHL3, Thorlabs), while the remaining polarimetric imaging system was kept unchanged. A filter wheel was added near the PSA side, allowing band-pass filters centred at 610, 650, and 690 nm (FBH610-10, FBH650-10 and FBH690-10, Thorlabs) to be inserted during acquisition. For each wavelength, Mueller matrix images were acquired using the same transmissive measurement configuration and processed using wavelength-specific calibrations These measurements were used as specialised external test data to evaluate the spectral generalisation of MMPD-Bench under wavelength polarimetric variations.

For all measurements, the corresponding reference polarimetric parameter maps were obtained using the Lu–Chipman decomposition procedure described in Appendix B.

## A.2. Data Processing

The resulting Mueller matrix images were partitioned into $256 \times 256 \times 16$ patches with a stride of 200 pixels. This standardised patch-based preparation enables controlled benchmarking of model performance while preserving the intrinsic microstructural heterogeneity between healthy and diseased bone-cell specimens.

The primary biological dataset contains 21,412 paired samples, where each sample consists of a Mueller matrix patch and its corresponding decomposition outputs. We split this dataset into approximately 80% training, 10% validation, and 10% testing subsets, while ensuring that both healthy and diseased bone-cell regions are represented in each subset.

In addition to the primary biological sets, we construct two specialised external test sets using the same partition rule. The first consists of waveplate measurements, including commercial half- and quarter-wave plates, and is used to assess cross-sample generalisation to retardance-dominant polarimetric responses. The second consists of multi-wavelength measurements acquired at 610, 650, and 690 nm, and is used to evaluate spectral generalisation under wavelength-dependent polarimetric variations.

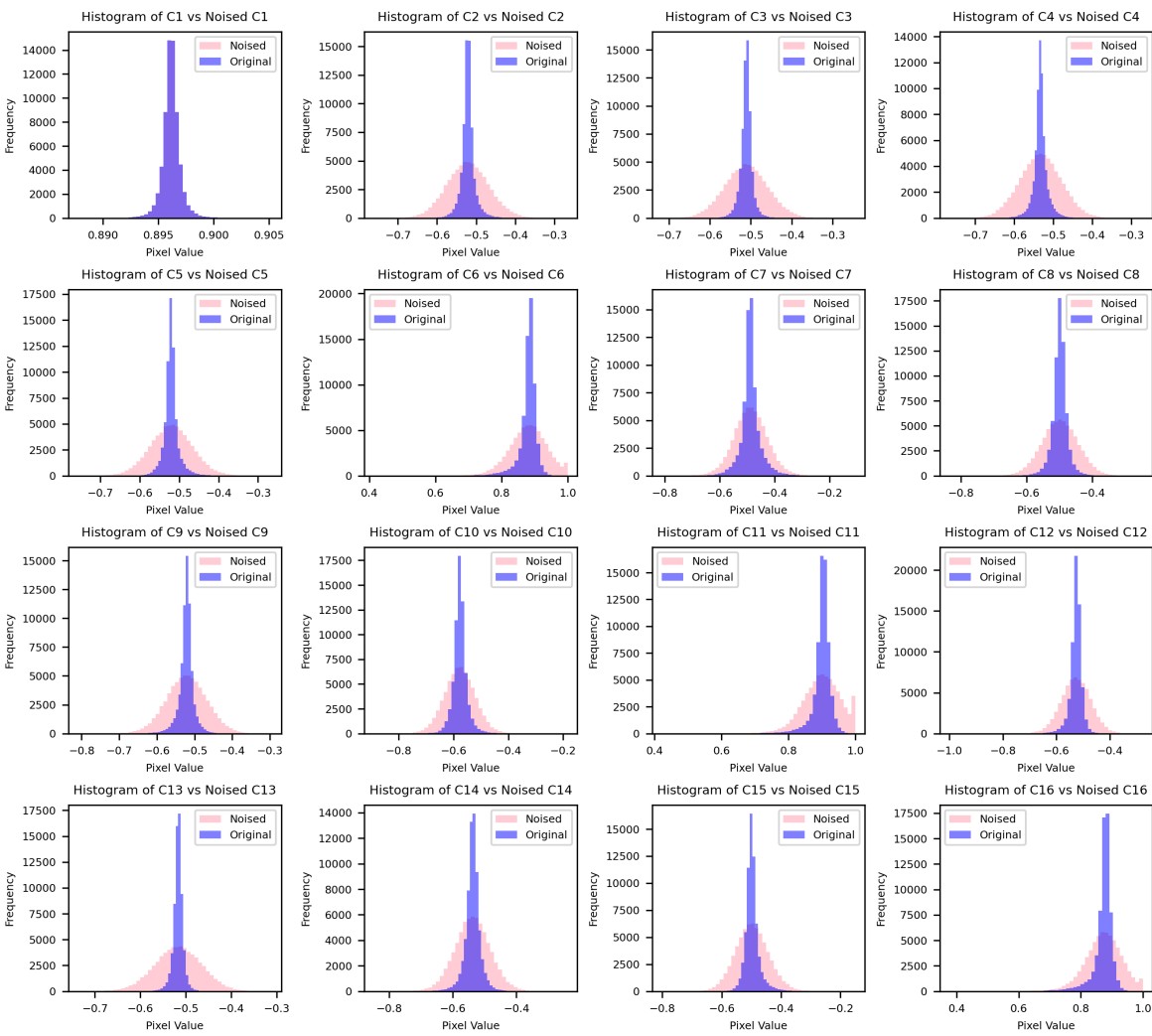

*Figure 5.* Histograms of pixel intensity distributions across a sampled 16-channel Mueller matrix normalized to [-1, 1]. The plots compare clean measurements against noisy observations perturbed by Gaussian noise, where the noise standard deviation is set to 10% of the pixel value standard deviation ($\sigma_{\text{noise}} = 0.1\sigma_{\text{pixel}}$).

## B. Mueller Matrix Polarization Decomposition

This section provides a concise and implementation-oriented summary of the Mueller matrix polar decomposition (MMPD) framework adopted throughout previous studies (Lu & Chipman, 1996; Zhang et al., 2025a). The purpose is not to rederive the formalism in full generality, but to specify the exact decomposition procedure, parameter definitions, and conventions used to generate the reference modalities in MMPD-Bench. This ensures reproducibility, unambiguous interpretation of the benchmark outputs, and consistent comparison between physics-based solvers and data-driven surrogates.

**Mueller matrix formation.** The polarisation state of a light beam can be described by the real-valued $4 \times 1$ Stokes vector $\mathbf{S} \in \mathbb{R}^4$. When a light beam with an initial polarization state $\mathbf{S}^{in}$ interacts with a sample, its polarization state is transformed described by a linear operator, the $4 \times 4$ real-valued Mueller Matrix, $\mathbf{M} \in \mathbb{R}^{4 \times 4}$. Then, the output state $\mathbf{S}^{out}$, is described by the mueller matrix transformation on $\mathbf{S}^{in}$ as

$$\underbrace{\begin{pmatrix} S_1^{out} \\ S_2^{out} \\ S_3^{out} \\ S_4^{out} \end{pmatrix}}_{\mathbf{S}^{out}} = \underbrace{\begin{pmatrix} M_{11} & M_{12} & M_{13} & M_{14} \\ M_{21} & M_{22} & M_{23} & M_{24} \\ M_{31} & M_{32} & M_{33} & M_{34} \\ M_{41} & M_{42} & M_{43} & M_{44} \end{pmatrix}}_{\mathbf{M}} \underbrace{\begin{pmatrix} S_1^{in} \\ S_2^{in} \\ S_3^{in} \\ S_4^{in} \end{pmatrix}}_{\mathbf{S}^{in}}. \tag{12}$$

**Lu–Chipman polar decomposition.** To obtain physically interpretable polarimetric parameters from each measured Mueller matrix, we adopt the polar decomposition that factors a physically realisable Mueller matrix into a diattenuator, a retarder, and a depolariser,

$$\mathbf{M} = \mathbf{M}_\Delta \mathbf{M}_R \mathbf{M}_D. \tag{13}$$

In our implementation, each matrix is first normalised by its intensity term, i.e. $\mathbf{M} \leftarrow \mathbf{M}/M_{11}$ (for $M_{11} \neq 0$), so that the subsequent parameter estimates are insensitive to overall gain.

**Extraction of diattenuation $\mathbf{M}_D$.** The diattenuation vector $\mathbf{D}$ is determined directly from the first column of the normalised matrix,

$$\mathbf{D} = \begin{pmatrix} \mathbf{M}_{12} \\ \mathbf{M}_{13} \\ \mathbf{M}_{14} \end{pmatrix}. \tag{14}$$

The magnitude of the diattenuation vector provides a scalar measure of the strength of diattenuation and is defined as

$$D = \|\mathbf{D}\|_2, \tag{15}$$

which is physically bounded to the interval $0 \leq D \leq 1$ for passive media. Then corresponding diattenuator $\mathbf{M}_D$ is constructed as

$$\mathbf{M}_D = \begin{pmatrix} 1 & \mathbf{D}^T \\ \mathbf{D} & \mathbf{m}_D \end{pmatrix}, \qquad \mathbf{m}_D = \sqrt{1 - D^2}\mathbf{I}_3 + \left(1 - \sqrt{1 - D^2}\right)\tilde{\mathbf{D}}\tilde{\mathbf{D}}^T, \tag{16}$$

where $\tilde{\mathbf{D}} = \mathbf{D}/D$ for $D \neq 0$ and $\mathbf{m}_D$ denotes the $3 \times 3$ submatrix which $\mathbf{m} = \mathbf{M}_{2:4, \, 2:4}$. Diattenuation is then removed via

$$\mathbf{M}' = \mathbf{M}\mathbf{M}_D^{-1}, \tag{17}$$

so that $\mathbf{M}'$ has zero diattenuation but, in general, contains both retardance $\mathbf{M}_R$ and depolarisation $\mathbf{M}_\Delta$.

**Separation of retardance $\mathbf{M}_R$ and depolarisation $\mathbf{M}_\Delta$** After removal of diattenuation, the Mueller matrix takes the form

$$\mathbf{M}' = \begin{pmatrix} 1 & \mathbf{0}^T \\ \mathbf{P}_\Delta & \mathbf{m}' \end{pmatrix}, \tag{18}$$

where the vector $\mathbf{P}_\Delta$ represents the polarizance associated with the depolarising component, describing the ability of the medium to generate a partially polarised output from unpolarised incident light, and the $3 \times 3$ submatrix $\mathbf{m}'$ contains the combined effects of retardance and depolarisation. Following the polar decomposition of real matrices, $\mathbf{m}'$ can be factorised as

$$\mathbf{m}' = \mathbf{m}_\Delta \mathbf{m}_R, \tag{19}$$

with $\mathbf{m}_\Delta$ symmetric positive semidefinite and $\mathbf{m}_R$ an orthogonal rotation matrix. Let $\lambda_1 \geq \lambda_2 \geq \lambda_3 \geq 0$ denote the eigenvalues of $\mathbf{m}'\mathbf{m}'^T$, with corresponding orthonormal eigenvectors $\hat{\mathbf{v}}_1, \hat{\mathbf{v}}_2, \hat{\mathbf{v}}_3$. For the non-singular case $(\det(\mathbf{m}') \neq 0)$, the depolarisation submatrix is given in closed form by

$$\mathbf{m}_\Delta = \pm \left[\mathbf{m}'\mathbf{m}'^T + \sum_{i<j} \sqrt{\lambda_i \lambda_j}\mathbf{I}\right]^{-1} \left[(\sqrt{\lambda_1} + \sqrt{\lambda_2} + \sqrt{\lambda_3})\mathbf{m}'\mathbf{m}'^T + \sqrt{\lambda_1 \lambda_2 \lambda_3}\mathbf{I}\right], \tag{20}$$

where the sign is chosen according to $\mathrm{sign}(\det(\mathbf{m}'))$ to ensure physical consistency. The retarder component then follows as

$$\mathbf{m}_R = \mathbf{m}_\Delta^{-1}\mathbf{m}'. \tag{21}$$

For the singular case $(\det(\mathbf{m}') = 0)$, $\mathbf{m}'$ admits the spectral representation

$$\mathbf{m}' = \sum_{i=1}^{3} \sqrt{\lambda_i}\,\hat{\mathbf{v}}_i\hat{\mathbf{u}}_i^T, \tag{22}$$

where $\hat{\mathbf{u}}_i$ denote the corresponding orthonormal eigenvectors of $\mathbf{m}'^T\mathbf{m}'$. Then the depolariser and retarder are constructed as

$$\mathbf{m}_\Delta = \pm\sum_{i=1}^{3} \sqrt{\lambda_i}\,\hat{\mathbf{v}}_i\hat{\mathbf{v}}_i^T, \qquad \mathbf{m}_R = \pm\sum_{i=1}^{3} \hat{\mathbf{v}}_i\hat{\mathbf{u}}_i^T, \tag{23}$$

corresponding to the minimum-retardance solution within the admissible family. Here the overall sign ambiguity reflects the invariance of the Mueller matrix under simultaneous sign changes of $\mathbf{m}_\Delta$ and $\mathbf{m}_R$, and is resolved by enforcing consistency with $\mathrm{sign}(\det(\mathbf{m}'))$, yielding the physically admissible minimum-retardance solution.

**Parameter extraction.** After obtaining $\mathbf{M}_\Delta$, $\mathbf{M}_R$, and $\mathbf{M}_D$, we compute a set of polarimetric parameters on a per-pixel basis: diattenuation magnitude $D$, depolarisation power $\Delta$, total retardance $R$, linear retardance $\delta$, optical rotation $\psi$, and fast-axis orientation $\theta$. The diattenuation magnitude is directly obtained from the diattenuation vector as

$$D = \|\mathbf{D}\|_2,$$

The depolarisation power is computed from the depolariser submatrix as

$$\Delta = 1 - \frac{|\mathrm{tr}(\mathbf{m}_\Delta)|}{3}, \tag{24}$$

and the total retardance is obtained from the retarder submatrix via

$$R = \cos^{-1}\left(\frac{\mathrm{tr}(\mathbf{m}_R) - 1}{2}\right), \tag{25}$$

with the argument clipped to $[-1, 1]$ for numerical stability. Optical rotation is estimated from the in-plane rotation of $\mathbf{m}_R$ as

$$\psi = \frac{1}{2}\tan^{-1}\left(\frac{\mathbf{m}_{R_{12}} - \mathbf{m}_{R_{21}}}{\mathbf{m}_{R_{11}} + \mathbf{m}_{R_{22}}}\right). \tag{26}$$

To separate circular and linear components, we form the circular-retarder submatrix $\mathbf{m}_{CR}(\psi)$ and recover the linear-retarder component $\mathbf{m}_{LR}$ by right-multiplication with $\mathbf{m}_{CR}^T$. The fast-axis orientation is then computed from the antisymmetric part of $\mathbf{m}_{LR}$ as

$$\theta = \frac{1}{2}\tan^{-1}\left(\frac{\mathbf{m}_{LR_{31}} - \mathbf{m}_{LR_{13}}}{\mathbf{m}_{LR_{23}} - \mathbf{m}_{LR_{32}}}\right), \tag{27}$$

and the linear retardance is computed following the standard relation used in polarimetry,

$$\eta = \cos^{-1}\left(\sqrt{(\mathbf{m}_{R_{11}} + \mathbf{m}_{R_{22}})^2 + (\mathbf{m}_{R_{21}} - \mathbf{m}_{R_{12}})^2} - 1\right). \tag{28}$$

These parameters yield spatially resolved maps used for downstream quantitative analysis and benchmarking.

# C. Related Works

## C.1. Deterministic Vision Models

These approaches treat MMPD as a deterministic inverse problem, learning a direct mapping from Mueller matrix observations to physical modalities. Early architectures based on Convolutional Neural Networks (CNNs) (LeCun & Bengio, 1998) successfully extracted local polarization textures, mapping specific channels to unimodal outputs like diattenuation and retardance (Li et al., 2017; Zhao et al., 2020; Ma et al., 2021; Yang et al., 2024). However, the inherent locality of convolutional inductive biases restricts the receptive field, preventing the capture of global physical continuity across large fields of view and leading to gradient instability in deeper networks (Tian et al., 2025; Chen et al., 2025; He et al., 2026).

To overcome these limitations, recent advancements leverage Vision Transformers (ViTs) and State-Space Models (SSMs) to model global dependencies and inter-channel physical correlations. ViTs utilize self-attention mechanisms to capture long-range interactions, adapting transformer architectures for high-dimensional spatial computing (Dosovitskiy et al., 2021; Liu et al., 2021a;b). Concurrently, SSMs exemplified by Vision-Mambas (Zhu et al., 2024; Liu et al., 2024b;a) as compelling alternatives, demonstrate the capacity of implicit sequence modeling to capture long-range dependencies in visual data. Unlike the quadratic complexity of ViTs, SSMs utilize selective scanning mechanisms to achieve linear computational complexity with respect to sequence length, enabling the efficient modeling of global physical dependencies in high-resolution scientific data without the heavy computational burden of global attention.

Despite these architectural innovations, these models are constrained by default pixel-level objectives, such as MSE and $L_1$ loss, suffer from *spectral bias* that models inherently behave low-frequency approximations (Rahaman et al., 2019; Xu et al., 2019). Standard mitigation strategies employ UNet-style skip connections to re-inject high-frequency details from multi-scale features (Yang et al., 2022; Liu et al., 2024a; Chen et al., 2025), but it lacks physical interpretability in adding the residuals as meaningful physical features, and risks in amplifying errors when given noised observation conditions. Another primary approach pivoted physics-inspired ViTs (Wei & Zhang, 2023; Li et al., 2023b; He et al., 2025b), which learn latent spectral representations of high-dimensional mappings governed by physical laws. To rigorously evaluate these approaches, we select SegMamba (Xing et al., 2025), SwinUMamba (Liu et al., 2024a), Swin Transformer (Liu et al., 2021b), and FactFormer (Li et al., 2023b) as representative benchmark deterministic vision models in this study.

## C.2. Probabilistic Generative Models

In contrast to regression-based methods which approximates the conditional mean of the target modalities, probabilistic generative models aim to learn the full joint distribution of the target modalities. Given the Mueller matrix observations, it transits MMPD from a deterministic mapping to a conditional generation task. Early attempts utilize Generative Adversarial Networks (GANs) to discriminate decomposed modalities from truth samples, where approaches in (Si et al., 2022; Yang et al., 2024) effectively constrain the decomposition process towards the target manifolds of polarimetric modalities. However, GANs often suffer from training instability due to the Minimax ojective, and easy to have the intrinsic mode collapse issue when the model fails to recover the full solution space, thereby limiting their ability to sufficiently represent the diverse polarization modalities.

Emerged as a robust alternative, Diffusion Models (DMs) constitute a powerful class of generative models capable of producing high-dimensional modality distributions, ranging from natural images to protein structures. By reversing a gradual noise-adding process, DMs learn to sample from the represented data manifold, exhibiting the gradual recovery from low-frequency global structure to high-frequency local details (Choi et al., 2022; Yang et al., 2023; Rissanen et al., 2022). Recent works (Moriconi et al., 2024; Zhang et al., 2025b; Zheng et al., 2025; Yang et al., 2026) have begun to adapt DMs for polarization feature translation, leveraging their iterative refinement process to decompose intricate topological structures or strategically sample and fuse features for downstream tasks, such as diseased zone segmentation, Hematoxylin-Eosin staining image synthesis. Crucially for scientific computing problems, this probabilistic formulation transforms MMPD from simple point-estimation to posterior sampling, theoretically enabling both uncertainty quantification and high-fidelity decomposition. In this study, we select conditional Denoising Diffusion Probabilistic Models (DDPM) (Ho et al., 2020) and Denoising Diffusion Implicit Models (DDIM) (Song et al., 2020a) as our generative benchmarks, assessing their unique capacity for uncertainty quantification against the regression baselines.

## C.3. Neural Operators

Apart from the deterministic and generative approaches typically adapted from computer vision, *Neural Operators* (NOs) offer a promising, physics-native alternative rooted in operator theory (Li et al., 2020; Kovachki et al., 2023; Azizzadenesheli et al., 2024; Cheng et al., 2025a; He et al., 2025b). Unlike standard deep learning models that learn mappings between finite-dimensional discrete vector spaces of patches and grids, NOs learn the underlying continuous operator between infinite-dimensional function spaces. By parameterizing the integral kernel in the spectral domain, spectral variants namely, Fourier Neural Operators(FNO), U-shaped FNO and multiwavelet-NO, have demonstrated exceptional success in solving scientific and inverse problems governed by differential Equations including fluid dynamics and wave propagation.

Despite the gap that no application of NOs in polarization modalities analysis are currently traced in the literature, we are motivated to introduce NOs in MMPD-bench by their physics analogy: the interaction of polarized light with observation specimens can be viewed as a continuous vectorial transformation field governed by the Stokes-Mueller calculus (Chipman et al., 2018). The task of MMPD naturally connects to the operator learning that maps the observed Mueller matrix field to the latent physical parameter fields. Furthermore, despite the success of NOs in dynamic forecasting, there remains a lack of theoretical and empirical validation of neural operators for the multimodal decomposition task from a single static modality. Adapting NOs to MMPD aims to investigate above gaps and benchmarking efficacy and scalability of NOs performance in this research field.

# D. Training Details.

We summarize the common training setup in Table 4, and report model-specific architecture and training configurations in Table 5; further implementation details are provided in Appendix H.

*Table 4.* Common training setup for baseline models.

| Setting | Configuration |
|---|---|
| Data setting | Clean and noisy training settings are considered separately |
| Input size | $256 \times 256$ |
| Optimizer | AdamW |
| Loss function | MAE for deterministic models; MSE for diffusion models forward to noise |
| Epochs | 100 |
| Initial LR | $2 \times 10^{-4}$ |
| LR Scheduler | Cosine Annealing with weight decay $10^{-4}$, $\eta_{\min} = 10^{-6}$ and $T_{\max} = 100$ |
| Hardware | `NVIDIA A100-SXM4-40GB` |

*Table 5.* Model architecture and specific training configurations for different baseline families.

| Family | Setting | Size (-s) | Size (-b) |
|---|---|---|---|
| **Mamba-based Models** | | | |
| **SU** | base_dim | 64 | 128 |
| | U-shape stages | 4 | 4 |
| | VSS blocks per stage | [2,2,6,2] | [2,2,12,2] |
| | d_conv | 4 | 4 |
| **SEG** | base_dim | 48 | 64 |
| | U-shape stages | 3 | 3 |
| | TriOrientedMamba blocks per stage | [2,2,2] | [2,4,2] |
| | kernel_size | (1,2,2) | (1,2,2) |
| **Transformer-based Models** | | | |
| **SU** | embed_dim | 96 | 128 |
| | U-shape stages | 4 | 4 |
| | Swin blocks per stage | [2,2,6,2] | [2,2,18,2] |
| | num_heads | [4,8,16,32] | [4,8,16,32] |
| | window_size | 8 | 8 |
| | drop_path_rate | 0.1 | 0.1 |
| **Fact-Former** | dim | 128 | 128 |
| | depth | 4 | 4 |
| | heads | 8 | 16 |
| | dim_head | 32 | 64 |
| | kernel_multiplier | 2 | 2 |
| | latent_multiplier | 1 | 1 |
| **Diffusion Models** | | | |
| **DDPM/ DDIM** | dim | 64 | 128 |
| | cf_guidance_scale | 7.5 | 7.5 |
| | cf_uncond_prob | 0.01 | 0.01 |
| | timesteps | 1000 | 1000 |
| **Neural Operators** | | | |
| **FNO/UNO** | spec_size | 32 | 64 |
| | embed_dim | 64 | 128 |
| | projection_dim | 64 | 128 |
| **UNO** | uno_out_channels | [32,64,64,64,32] | [64,128,128,128,64] |
| | uno_n_modes | [[8,8],[8,8],[4,4],[8,8],[8,8]] | [[16,16],[16,16],[8,8],[16,16],[16,16]] |
| | uno_depth | 5 | 5 |

## E. Computation Costs

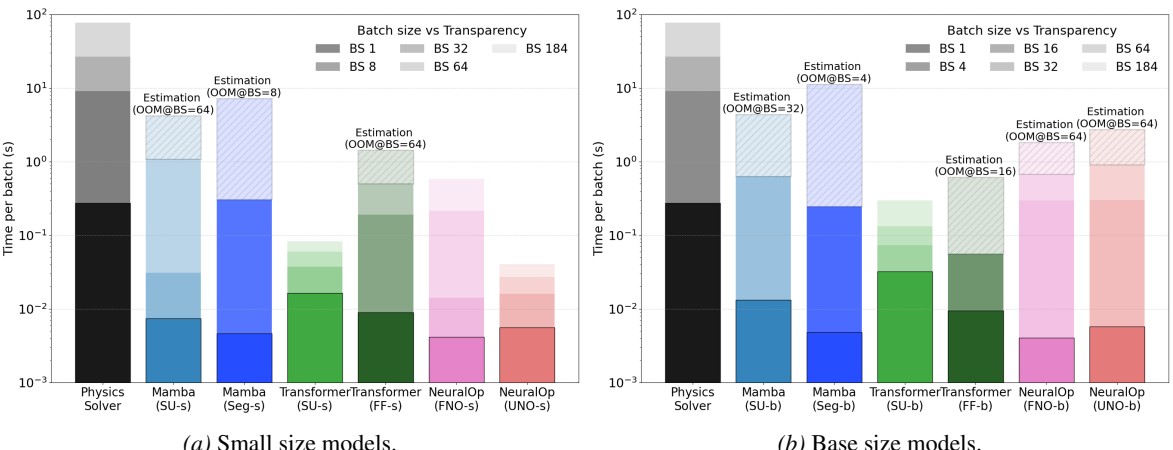

*(a)* Small size models.   *(b)* Base size models.

*Figure 6.* Runtime Efficiency and Scalability in inference. Log-scale comparison of wall-clock inference time per batch across increasing batch sizes (BS) for (a) Small- and (b) Base- size model configurations on one `NVIDIA A100-SXM4-40GB`. The conventional Physics Solver exhibits steep linear scaling, showing the computational bottleneck of numerical decomposition. In contrast, PiVD surrogates demonstrate orders-of-magnitude acceleration. Notably, FNOs achieve the highest inference efficiency in $BS = 1$, and SwinU Transfromers strong scalability and inferencing efficiency when BS grows. Whilst Factformers and Mamba architectures encounter Out-of-Memory (OOM) constraints at larger batch sizes.

*Table 6.* Computational costs of baseline and proposed MMPD models on polarization modality prediction. We report the model performance in terms of parameter count, memory usage, and floating-point operations (FLOPs) per forward pass. The results are gathered given model configurations on one `NVIDIA A100-SXM4-40GB` with sample `batch_size = 1`.

| Family | Models | Parameter Count | Memory Usage | FLOPs per Forward Pass |
|---|---|---|---|---|
| Mamba | SU-s | 27.87 M | 32.06 MB | 27.87 MFLOPs |
| | SU-b | 169.25 M | 173.44 MB | 169.25 MFLOPs |
| | Seg-s | 2.52 M | 6.17 MB | 2.52 MFLOPs |
| | Seg-b | 5.17 M | 9.36 MB | 5.17 MFLOPs |
| Transformer | SU-s | 41.29 M | 45.49 MB | 41.29 MFLOPs |
| | SU-b | 149.04 M | 153.23 MB | 149.04 MFLOPs |
| | FF-s | 2.09 M | 6.28 MB | 2.08 MFLOPs |
| | FF-b | 6.03 M | 10.23 MB | 6.02 MFLOPs |
| Diffusion | DDIM-s | 57.78 M | 231.10 MB | 57.78 MFLOPs |
| | DDIM-b | 230.83 M | 923.31 MB | 230.83 MFLOPs |
| | DDPM-s | 57.74 M | 230.95 MB | 57.74 MFLOPs |
| | DDPM-b | 230.68 M | 922.72 MB | 230.68 MFLOPs |
| NeuralOp | FNO-s | 3.92 M | 8.11 MB | 2.24 MFLOPs |
| | FNO-b | 242.52 M | 246.72 MB | 138.62 MFLOPs |
| | UNO-s | 7.46 M | 49.48 MB | 7.52 MFLOPs |
| | UNO-b | 97.84 M | 139.84 MB | 97.8 MFLOPs |

# F. Evaluation Metrics

Let $\Omega$ denote the evaluation mask with $|\Omega|$ pixels. For each modality $u$, let $u(p)$ be the ground-truth value and $\hat{u}(p)$ the predicted value at pixel $p \in \Omega$.

## F.1. Vision-level measurements as essential metrics

**PSNR.** Let $\mathrm{MSE}(u, \hat{u}) = \frac{1}{|\Omega|} \sum_{p \in \Omega}(u(p) - \hat{u}(p))^2$. Then $\mathrm{PSNR} = 10 \log_{10}\left(\frac{\mathrm{MAX}^2}{\mathrm{MSE}}\right)$.

**SSIM.** SSIM is computed over local windows and averaged: $\mathrm{SSIM}(u, \hat{u}) = \frac{(2\mu_u \mu_{\hat{u}} + c_1)(2\sigma_{u\hat{u}} + c_2)}{(\mu_u^2 + \mu_{\hat{u}}^2 + c_1)(\sigma_u^2 + \sigma_{\hat{u}}^2 + c_2)}$.

## F.2. Physics-level consistency as references

**Scale-normalized parameter error.** For scalar modalities $u \in \{D, \Delta, \theta, \psi, \eta, R\}$ we report $\mathrm{nMAE}(u) = \frac{1}{|\Omega|} \sum_{p \in \Omega} \frac{|u(p) - \hat{u}(p)|}{s_u}$, with $s_u$ chosen from known modality ranges and the standard deviations.

**Wasserstein-1 distance.** For a scalar map $u$, treat $\{u(p)\}_{p \in \Omega}$ as samples. The 1-Wasserstein distance is $W_1(P, Q) = \int |F_P(t) - F_Q(t)| \, dt$. In 1D with equal-sized samples, $W_1$ is estimated by sorting and averaging absolute differences.

**Retardance consistency.** Let $\rho$ denote the retardance vector representation of the retarder component. With linear and circular components corresponding to orthogonal subcomponents, total retardance satisfies $R = \|\rho\|_2 = \sqrt{\rho_L^2 + \rho_C^2}$. When $\eta$ and $\psi$ parameterize the magnitudes of linear and circular components, we penalize violations via $e_{\mathrm{ret}} = \frac{1}{|\Omega|} \sum_{p \in \Omega} (\hat{R}(p)^2 - \hat{\eta}(p)^2 - \hat{\psi}(p)^2)^2$. We additionally report bound-violation rates for parameters with known admissible intervals.

## F.3. Params Range Value

In both phyics decomposition and model generation, the polarimetric parameters' value range in the physical space is provided in Table 7.

*Table 7.* Polarimetric parameters range.

| Parameter | Symbol | Range |
|---|---|---|
| Diattenuation | $D$ | $[0, 1]$ |
| Depolarisation | $\Delta$ | $[0, 1]$ |
| Linear Retardance | $\eta$ | $[0, \pi)$ |
| Fast Axis Orientation | $\theta$ | $[-\pi/2, \pi/2)$ |
| Optical Rotation | $\psi$ | $[-\pi/2, \pi/2)$ |
| Total Retardance | $R$ | $[0, \pi)$ |

# G. MMPD Results

The polarimetric datasets used in this appendix are representative of practical wide-field transmission polarimetry on thin biological sections, where the measured Mueller matrices reflect a mixture of ordered anisotropic microstructure and heterogeneous multiple scattering. In particular, healthy regions typically exhibit more organised fibrillar features that manifest as spatially coherent linear retardance and oriented fast-axis maps, whereas diseased regions tend to introduce stronger depolarisation and disrupted anisotropy, yielding reduced orientation coherence and increased spatial variability (Zhao et al., 2025; He et al., 2025a). This combination makes data physically rich, mirroring the conditions where polarimetric decomposition is most needed but ground-truth parameters are generally unavailable.

In these thin-section transmission measurements, the dominant sample contrasts are linear retardance $\eta$, fast-axis orientation $\theta$ and depolarisation $\Delta$, while the apparent optical rotation $\psi$ is expected to be weak and is more susceptible to residual system errors and noise amplification in polar decomposition (Zhao et al., 2025; He et al., 2025a). For this reason, we report $\psi$ as a fixed reference value in the summary tables, and focus the quantitative comparisons on parameters that are robustly supported by the data (e.g., $D$, $\Delta$, $\eta$, and $\theta$). Likewise, we present $R$ as a fixed reference because, under negligible circular contribution, the total retardance $R$ provides limited additional information beyond the linear retardance $\eta$ for these specimens. This choice avoids over-interpreting a weakly identifiable degree of freedom and keeps the benchmark aligned with practical tissue polarimetry, where decomposition is primarily used to quantify linear birefringence-related contrast and depolarising behaviour.

### G.1. Results on clear MM

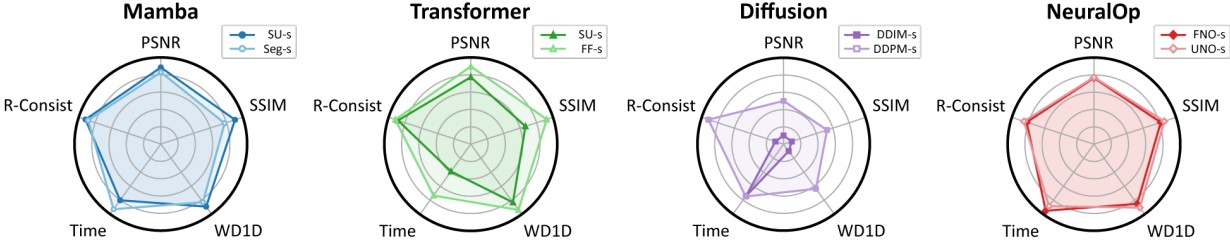

*Figure 7.* Multi-dimensional performance analysis across modalities for models (-s) against 5 dimensions of metrics on clear observations - 1. PSNR ↑: Pixel-wise reconstruction fidelity and signal quality across modalities; 2. SSIM ↑: Preservation of structural information and spatial patterns; 3. WD1D ↓: Global statistical alignment across the entire test population; 4.Time ↓: runtime to inference per sample (for diffusion models, this plot compares the runtime to denoise one step.); 5. R-Consist ↑: consistency to the physical retardance.

*Table 8.* Quantitative benchmarks for polarimetric decomposition generation quality on combined bone cells. The evaluation measures pixel accuracy and structural similarity with **clear** Mueller matrix observation decompositions using PSNR and SSIM metrics. Results are reported as (mean ± std) with color coding indicating performance rankings: the *top three* models for each modality are highlighted in (1st) , (2nd) , and (3rd) . The worst performance in each column is underlined.

| **Models** | **ImageD** | **ImageDelta** | **ImageEta** | **ImageTheta** | **ImagePsi** | **ImageR** |
|---|---|---|---|---|---|---|
| | | | **PSNR ↑ (dB)** | | | |
| Mamba (SU-s) | 45.9 ± 1.8 | 53.3 ± 3.8 | 45.1 ± 2.8 | 18.8 ± 5.0 | 51.4 ± 1.6 | 47.0 ± 3.0 |
| Mamba (SU-b) | 48.8 ± 2.0 | 55.0 ± 3.0 | 44.1 ± 2.6 | 24.8 ± 5.4 | 51.1 ± 1.5 | 47.8 ± 3.1 |
| Mamba (Seg-s) | 44.5 ± 3.3 | 52.5 ± 3.0 | 42.0 ± 2.6 | 15.5 ± 4.9 | 49.5 ± 1.4 | 47.7 ± 3.0 |
| Mamba (Seg-b) | 44.7 ± 3.6 | 53.2 ± 3.5 | 43.7 ± 2.9 | 15.6 ± 4.9 | 50.1 ± 1.5 | 47.2 ± 3.3 |
| Transformer (SU-s) | 40.7 ± 2.9 | 47.0 ± 5.5 | 43.2 ± 2.5 | 14.1 ± 4.7 | 50.1 ± 1.6 | 46.4 ± 2.9 |
| Transformer (SU-b) | 43.1 ± 2.3 | 49.6 ± 4.3 | 43.4 ± 2.3 | 15.2 ± 4.8 | 50.5 ± 1.6 | 46.8 ± 2.5 |
| Transformer (FF-s) | 43.6 ± 2.5 | 53.3 ± 3.8 | 44.2 ± 2.6 | 23.8 ± 4.3 | 51.1 ± 1.6 | 47.6 ± 3.2 |
| Transformer (FF-b) | 47.7 ± 1.7 | 57.7 ± 4.3 | 43.0 ± 2.1 | 24.1 ± 5.0 | 52.5 ± 1.6 | 48.1 ± 2.6 |
| Diffusion (DDIM-s) | 17.9 ± 4.9 | 19.9 ± 5.4 | 20.6 ± 5.8 | 14.8 ± 4.1 | 28.3 ± 3.6 | 20.3 ± 5.7 |
| Diffusion (DDIM-b) | 24.1 ± 6.7 | 25.7 ± 7.0 | 24.1 ± 5.6 | 18.8 ± 3.6 | 30.8 ± 3.5 | 25.4 ± 5.9 |
| Diffusion (DDPM-s) | 31.9 ± 5.6 | 53.3 ± 3.8 | 30.1 ± 7.2 | 16.0 ± 2.5 | 32.1 ± 4.6 | 29.7 ± 6.1 |
| Diffusion (DDPM-b) | 28.9 ± 7.4 | 28.2 ± 7.7 | 29.4 ± 7.4 | 20.0 ± 4.1 | 51.1 ± 1.5 | 30.0 ± 7.7 |
| NeuralOp (FNO-s) | 40.0 ± 3.1 | 44.0 ± 6.2 | 42.9 ± 2.2 | 19.7 ± 3.1 | 48.6 ± 2.1 | 46.2 ± 3.3 |
| NeuralOp (FNO-b) | 41.6 ± 2.8 | 51.3 ± 5.0 | 43.7 ± 2.4 | 25.9 ± 4.5 | 51.3 ± 1.7 | 47.9 ± 3.0 |
| NeuralOp (UNO-s) | 39.4 ± 3.2 | 42.6 ± 6.3 | 42.4 ± 2.4 | 24.5 ± 4.1 | 46.3 ± 2.5 | 46.5 ± 3.0 |
| NeuralOp (UNO-b) | 40.2 ± 3.0 | 50.1 ± 3.8 | 42.8 ± 2.4 | 27.3 ± 4.4 | 51.7 ± 1.9 | 47.4 ± 2.9 |
| | | | **SSIM ↑ (%)** | | | |
| Mamba (SU-s) | 97.0 ± 1.0 | 99.0 ± 1.0 | 85.0 ± 9.0 | 86.0 ± 7.0 | 99.0 ± 3.0 | 92.0 ± 5.0 |
| Mamba (SU-b) | 98.2 ± 0.6 | 99.3 ± 0.6 | 82.5 ± 9.8 | 96.5 ± 2.4 | 99.6 ± 0.3 | 93.0 ± 4.9 |
| Mamba (Seg-s) | 94.6 ± 2.5 | 98.2 ± 1.6 | 75.8 ± 11.2 | 64.0 ± 15.0 | 99.6 ± 0.6 | 95.3 ± 3.0 |
| Mamba (Seg-b) | 96.2 ± 2.0 | 99.0 ± 1.2 | 81.6 ± 9.9 | 64.1 ± 14.6 | 99.6 ± 0.6 | 93.9 ± 4.5 |
| Transformer (SU-s) | 89.8 ± 3.1 | 97.9 ± 1.8 | 81.1 ± 9.6 | 39.5 ± 18.0 | 99.4 ± 0.6 | 93.1 ± 4.2 |
| Transformer (SU-b) | 94.1 ± 1.7 | 98.7 ± 1.0 | 81.6 ± 9.8 | 54.7 ± 15.9 | 99.4 ± 0.6 | 93.7 ± 4.0 |
| Transformer (FF-s) | 94.5 ± 1.9 | 99.2 ± 0.7 | 82.2 ± 9.7 | 95.2 ± 2.0 | 99.6 ± 0.4 | 92.2 ± 5.3 |
| Transformer (FF-b) | 97.2 ± 0.7 | 99.8 ± 0.3 | 79.6 ± 10.4 | 95.9 ± 2.0 | 99.6 ± 0.4 | 94.4 ± 3.7 |
| Diffusion (DDIM-s) | 42.7 ± 16.1 | 54.6 ± 18.2 | 61.5 ± 11.3 | 73.7 ± 11.1 | 70.4 ± 7.3 | 61.5 ± 11.5 |
| Diffusion (DDIM-b) | 58.9 ± 12.1 | 85.9 ± 8.6 | 52.4 ± 7.8 | 85.0 ± 6.3 | 78.3 ± 6.1 | 68.4 ± 9.0 |
| Diffusion (DDPM-s) | 79.6 ± 7.2 | 80.8 ± 9.9 | 75.0 ± 13.6 | 71.3 ± 14.4 | 87.4 ± 6.5 | 73.8 ± 11.2 |
| Diffusion (DDPM-b) | 80.1 ± 12.3 | 79.4 ± 9.4 | 80.8 ± 11.1 | 87.9 ± 7.5 | 89.9 ± 7.0 | 84.0 ± 9.8 |
| NeuralOp (FNO-s) | 88.6 ± 4.1 | 94.9 ± 4.6 | 81.9 ± 7.1 | 81.3 ± 7.8 | 99.3 ± 0.7 | 93.2 ± 4.3 |
| NeuralOp (FNO-b) | 91.6 ± 3.3 | 98.9 ± 1.6 | 81.1 ± 9.9 | 97.1 ± 1.9 | 99.6 ± 0.5 | 93.9 ± 4.1 |
| NeuralOp (UNO-s) | 87.0 ± 4.6 | 92.4 ± 5.4 | 80.3 ± 8.4 | 94.0 ± 2.6 | 99.2 ± 0.8 | 93.8 ± 3.9 |
| NeuralOp (UNO-b) | 89.1 ± 4.2 | 97.4 ± 3.5 | 78.4 ± 10.8 | 97.4 ± 1.3 | 99.5 ± 0.7 | 95.2 ± 2.9 |

*Table 9.* Quantitative benchmarks for polarimetric decomposition generation quality on combined bone cells. The evaluation measures statistical distance with **clear** Mueller matrix observation decompositions using WD-1D metrics. Results are reported as (mean ± st) for WD-1D per sample, and mean for WD-1D over the test set. The color coding indicates performance rankings: the *top three* models for each modality are highlighted in (1st), (2nd), and (3rd). The worst performance in each column is underlined.

| Models | ImageD | ImageDelta | ImageEta | ImageTheta | ImagePsi | ImageR |
|---|---|---|---|---|---|---|
| **WD-1d ↓ (per sample)** | | | | | | |
| Mamba (SU-s) | 0.34 ± 0.15 | 0.18 ± 0.13 | 1.20 ± 0.46 | 6.76 ± 3.97 | 0.45 ± 0.11 | 0.79 ± 0.41 |
| Mamba (SU-b) | 0.38 ± 0.07 | 0.20 ± 0.02 | 2.01 ± 0.08 | 0.46 ± 0.12 | 0.42 ± 0.04 | 1.25 ± 0.07 |
| Mamba (Seg-s) | 0.81 ± 0.54 | 0.31 ± 0.34 | 1.90 ± 0.90 | 11.03 ± 7.72 | 0.71 ± 0.18 | 0.58 ± 0.89 |
| Mamba (Seg-b) | 0.43 ± 0.53 | 0.22 ± 0.39 | 1.43 ± 1.02 | 12.3 ± 7.91 | 0.62 ± 0.17 | 0.75 ± 1.06 |
| Transformer (SU-s) | 1.09 ± 0.40 | 0.22 ± 0.38 | 1.50 ± 0.94 | 11.34 ± 7.27 | 0.54 ± 0.25 | 0.67 ± 0.96 |
| Transformer (SU-b) | 0.56 ± 0.23 | 0.19 ± 0.11 | 1.45 ± 0.45 | 7.49 ± 4.37 | 0.45 ± 0.14 | 0.61 ± 0.37 |
| Transformer (FF-s) | 0.54 ± 0.24 | 0.16 ± 0.15 | 1.39 ± 0.45 | 2.58 ± 1.11 | 0.49 ± 0.14 | 0.72 ± 0.40 |
| Transformer (FF-b) | 0.60 ± 0.14 | 0.05 ± 0.10 | 1.65 ± 0.40 | 1.88 ± 1.20 | 0.35 ± 0.11 | 0.60 ± 0.32 |
| Diffusion (DDIM-s) | 19.9 ± 16.4 | 11.1 ± 10.2 | 9.46 ± 10.5 | 23.4 ± 14.6 | 5.55 ± 3.15 | 10.4 ± 10.7 |
| Diffusion (DDIM-b) | 7.67 ± 9.48 | 3.87 ± 6.64 | 8.40 ± 10.5 | 7.75 ± 3.53 | 3.97 ± 2.86 | 6.36 ± 9.51 |
| Diffusion (DDPM-s) | 2.40 ± 1.81 | 2.97 ± 2.42 | 4.80 ± 6.13 | 13.8 ± 7.55 | 3.60 ± 2.08 | 4.72 ± 5.07 |
| Diffusion (DDPM-b) | 4.78 ± 10.1 | 4.34 ± 10.2 | 4.94 ± 10.8 | 6.11 ± 4.92 | 3.67 ± 5.90 | 4.79 ± 9.84 |
| NeuralOp (FNO-s) | 1.23 ± 0.76 | 0.58 ± 0.69 | 1.55 ± 1.11 | 8.67 ± 4.03 | 0.58 ± 0.32 | 0.73 ± 1.21 |
| NeuralOp (FNO-b) | 0.84 ± 0.32 | 0.15 ± 0.17 | 1.50 ± 0.45 | 1.84 ± 0.91 | 0.45 ± 0.14 | 0.62 ± 0.33 |
| NeuralOp (UNO-s) | 1.30 ± 0.54 | 0.67 ± 0.50 | 1.75 ± 0.67 | 3.25 ± 1.64 | 0.93 ± 0.39 | 0.74 ± 0.72 |
| NeuralOp (UNO-b) | 1.12 ± 0.48 | 0.33 ± 0.40 | 1.69 ± 0.53 | 1.93 ± 1.03 | 0.39 ± 0.16 | 0.61 ± 0.53 |
| **WD-1d ↓ (whole test set)** | | | | | | |
| Mamba (SU-s) | 0.29 | 0.17 | 1.19 | 6.54 | 0.45 | 0.78 |
| Mamba (SU-b) | 0.28 | 0.15 | 1.42 | 1.60 | 0.50 | 0.55 |
| Mamba (Seg-s) | 0.73 | 0.29 | 1.89 | 10.6 | 0.71 | 0.42 |
| Mamba (Seg-b) | 0.39 | 0.21 | 1.41 | 9.67 | 0.62 | 0.63 |
| Transformer (SU-s) | 0.93 | 0.19 | 1.49 | 9.42 | 0.54 | 0.60 |
| Transformer (SU-b) | 0.47 | 0.07 | 1.45 | 5.78 | 0.44 | 0.54 |
| Transformer (FF-s) | 0.45 | 0.08 | 1.39 | 2.25 | 0.48 | 0.64 |
| Transformer (FF-b) | 0.40 | 0.03 | 1.65 | 1.70 | 0.35 | 0.54 |
| Diffusion (DDIM-s) | 19.8 | 11.1 | 9.34 | 22.9 | 5.55 | 10.2 |
| Diffusion (DDIM-b) | 7.53 | 3.80 | 8.39 | 6.45 | 3.97 | 6.25 |
| Diffusion (DDPM-s) | 2.18 | 2.88 | 4.59 | 6.30 | 3.58 | 4.49 |
| Diffusion (DDPM-b) | 4.57 | 4.34 | 4.85 | 4.25 | 3.67 | 4.71 |
| NeuralOp (FNO-s) | 0.87 | 0.36 | 1.49 | 6.95 | 0.54 | 0.61 |
| NeuralOp (FNO-b) | 0.73 | 0.06 | 1.50 | 1.55 | 0.45 | 0.57 |
| NeuralOp (UNO-s) | 1.09 | 0.54 | 1.73 | 3.00 | 0.91 | 0.64 |
| NeuralOp (UNO-b) | 0.95 | 0.21 | 1.68 | 1.73 | 0.38 | 0.51 |

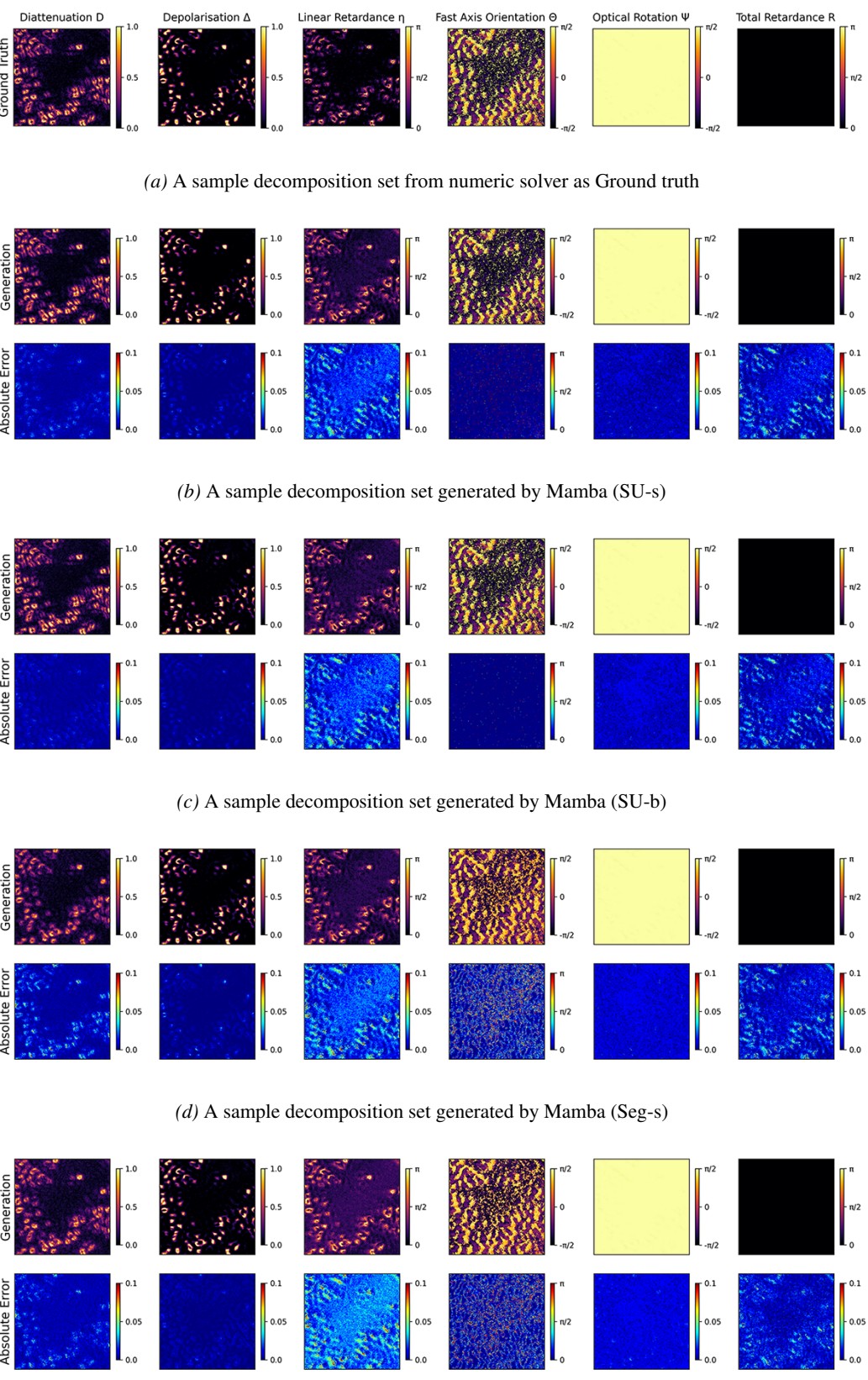

Figure 8. A. Qualitative evaluation of polarimetric parameter estimation by **Mamba**-based models.

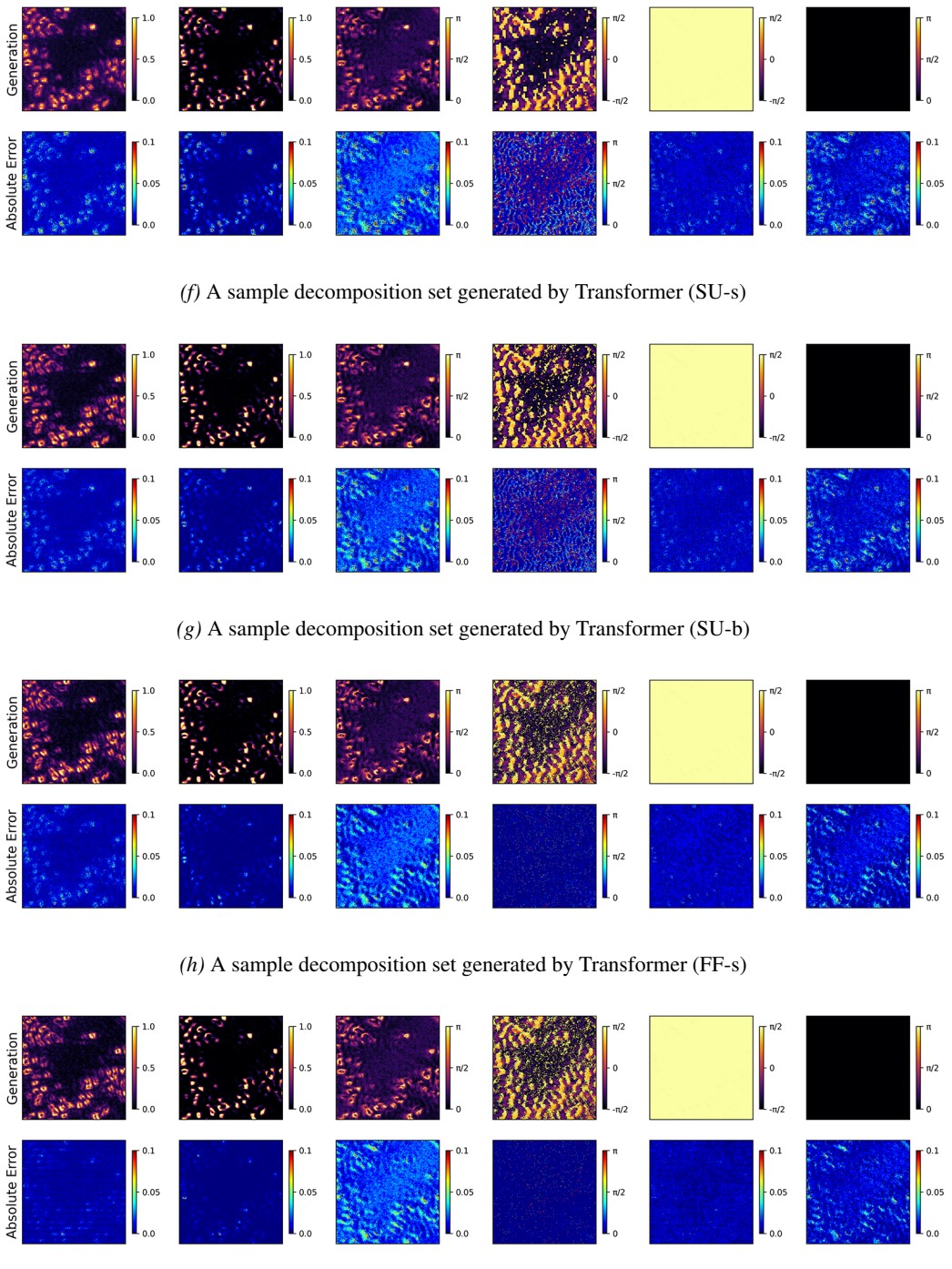

*(f)* A sample decomposition set generated by Transformer (SU-s)

*(g)* A sample decomposition set generated by Transformer (SU-b)

*(h)* A sample decomposition set generated by Transformer (FF-s)

*(i)* A sample decomposition set generated by Transformer (FF-b)

*Figure 8.* B. Qualitative evaluation of polarimetric parameter estimation by **Transformer**-based models.

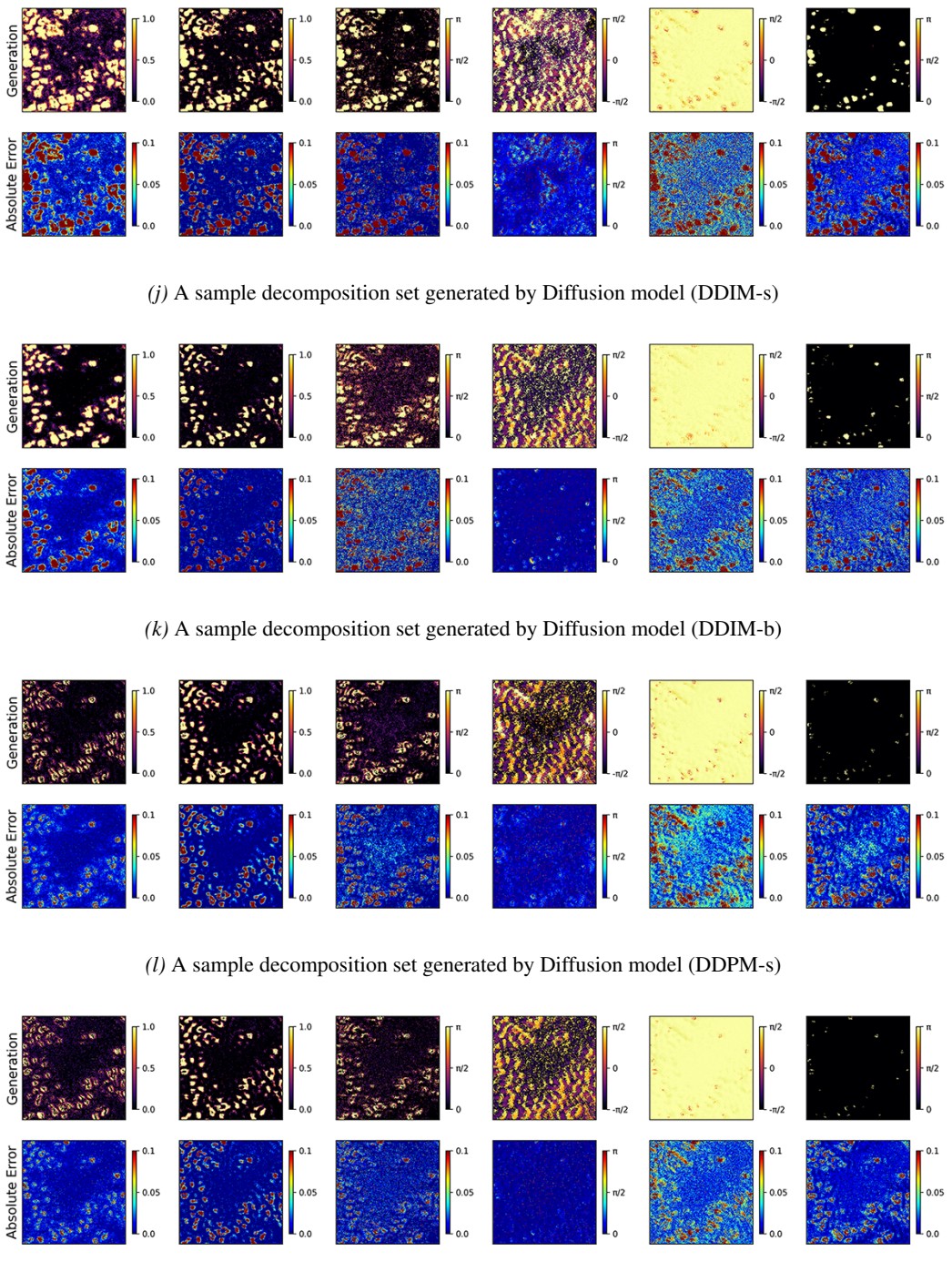

*(j)* A sample decomposition set generated by Diffusion model (DDIM-s)

*(k)* A sample decomposition set generated by Diffusion model (DDIM-b)

*(l)* A sample decomposition set generated by Diffusion model (DDPM-s)

*(m)* A sample decomposition set generated by Diffusion model (DDPM-b)

*Figure 8.* C. Qualitative evaluation of polarimetric parameter estimation by **Diffusion**-based models.

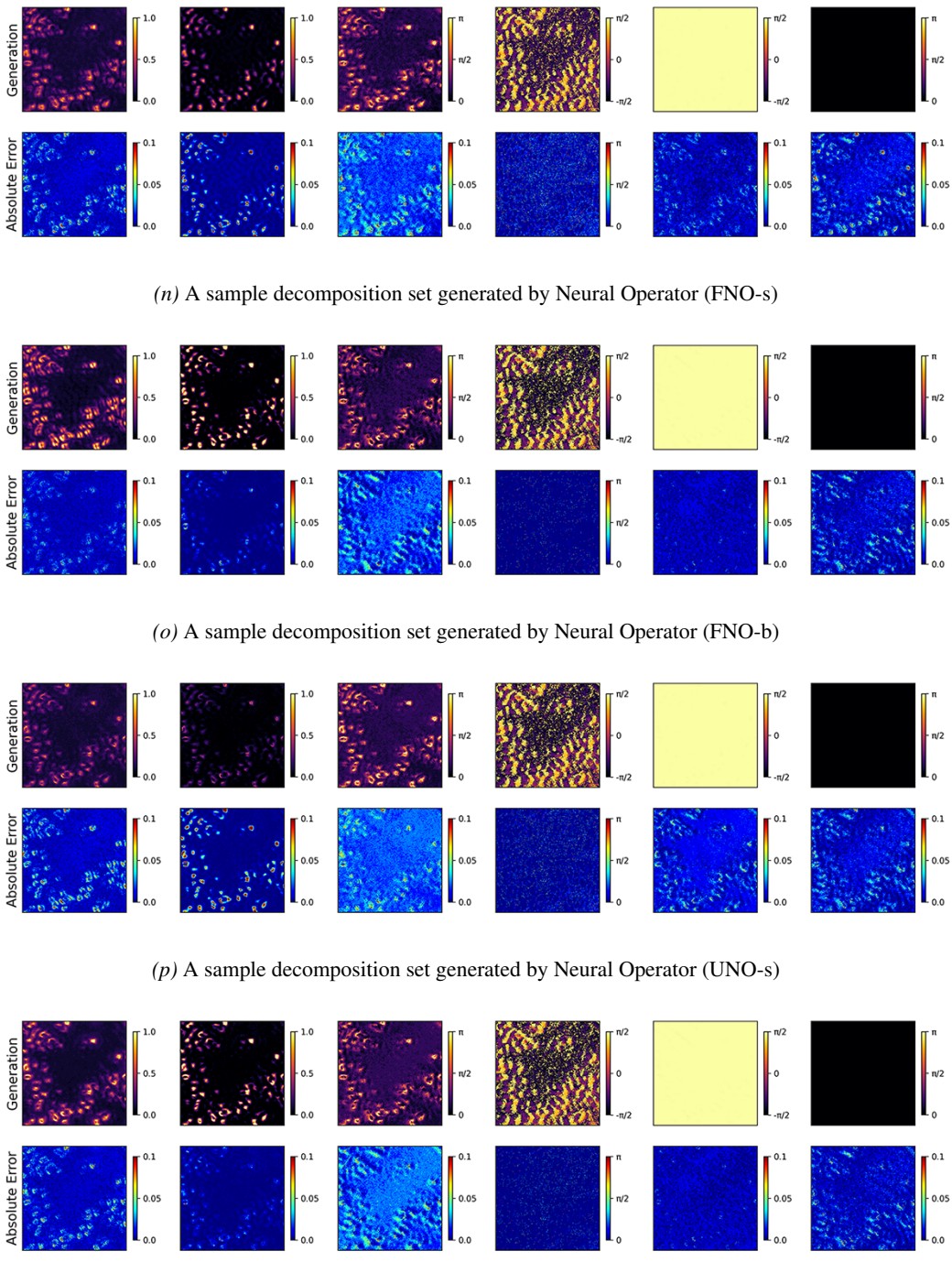

*(n)* A sample decomposition set generated by Neural Operator (FNO-s)

*(o)* A sample decomposition set generated by Neural Operator (FNO-b)

*(p)* A sample decomposition set generated by Neural Operator (UNO-s)

*(q)* A sample decomposition set generated by Neural Operator (UNO-b)

*Figure 8.* D. Qualitative evaluation of polarimetric parameter estimation by **Neural Operator**-based models. The figure presents a visual comparison across six polarimetric decomposition parameters: Diattenuation (D), Depolarization ($\Delta$), Linear Retardance ($\eta$), Fast Axis Orientation ($\theta$), Optical Rotation ($\psi$), and Total Retardance (R). The rows display the Ground Truth (top), the model's Prediction (1st row of each subfigure), and the corresponding Absolute Error map (2nd row of each subfigure). All predicted values have been remapped to their physical ranges for accurate comparison. The color bars on the right of each map indicate the value scale for that specific parameter and its error.

## G.2. Results on noised MM (Table position compiled mismatch)

*Table 10.* Robustness benchmarks on combined bone cells under Gaussian **noise** perturbation ($\sigma = 0.05$). The models were trained and evaluated on noisy Mueller matrix inputs to simulate experimental measurement errors. Results are reported as (mean $\pm$ std) for PSNR and SSIM. Color coding indicates performance rankings: the *top three* models for each modality are highlighted in (1st) , (2nd) , and (3rd) . The worst performance in each column is underlined.

| Models | ImageD | ImageDelta | ImageEta | ImageTheta | ImagePsi | ImageR |
|---|---|---|---|---|---|---|
| | | | **PSNR ↑ (dB)** | | | |
| Mamba (SU-s) | $43.0 \pm 1.8$ | $50.4 \pm 3.6$ | $43.6 \pm 2.1$ | $13.9 \pm 4.7$ | $51.1 \pm 1.8$ | $46.9 \pm 2.0$ |
| Mamba (SU-b) | $43.2 \pm 1.7$ | $50.7 \pm 3.2$ | $44.0 \pm 2.1$ | $14.1 \pm 4.7$ | $51.1 \pm 1.7$ | $46.9 \pm 2.2$ |
| Mamba (Seg-s) | $41.5 \pm 2.7$ | $46.5 \pm 5.4$ | $43.4 \pm 2.4$ | $13.4 \pm 4.5$ | $49.6 \pm 1.6$ | $46.1 \pm 2.7$ |
| Mamba (Seg-b) | $41.3 \pm 2.6$ | $47.1 \pm 4.8$ | $41.6 \pm 2.3$ | $13.3 \pm 4.5$ | $48.8 \pm 1.4$ | $46.6 \pm 2.7$ |
| Transformer (SU-s) | $40.3 \pm 3.3$ | $45.3 \pm 5.9$ | $43.3 \pm 2.2$ | $13.9 \pm 4.6$ | $49.7 \pm 1.7$ | $45.9 \pm 2.7$ |
| Transformer (SU-b) | $41.9 \pm 2.8$ | $48.0 \pm 5.2$ | $43.6 \pm 2.3$ | $14.2 \pm 4.7$ | $50.1 \pm 1.6$ | $46.3 \pm 2.6$ |
| Transformer (FF-s) | $43.3 \pm 2.0$ | $50.1 \pm 3.7$ | $43.9 \pm 2.4$ | $14.4 \pm 4.7$ | $50.3 \pm 1.5$ | $46.7 \pm 2.6$ |
| Transformer (FF-b) | $43.7 \pm 2.0$ | $50.9 \pm 3.8$ | $44.6 \pm 2.5$ | $14.3 \pm 4.7$ | $50.0 \pm 1.4$ | $46.6 \pm 2.7$ |
| Diffusion (DDIM-s) | $32.6 \pm 6.0$ | $34.3 \pm 5.7$ | $34.8 \pm 7.2$ | $11.3 \pm 4.1$ | $31.3 \pm 5.3$ | $33.8 \pm 6.6$ |
| Diffusion (DDIM-b) | $35.0 \pm 4.8$ | $38.6 \pm 6.8$ | $37.8 \pm 5.8$ | $10.7 \pm 3.9$ | $40.2 \pm 3.7$ | $36.9 \pm 5.3$ |
| Diffusion (DDPM-s) | $33.6 \pm 6.7$ | $33.7 \pm 4.5$ | $35.8 \pm 5.5$ | $10.5 \pm 3.7$ | $35.2 \pm 4.0$ | $35.2 \pm 5.2$ |
| Diffusion (DDPM-b) | $35.1 \pm 3.9$ | $34.2 \pm 3.6$ | $38.5 \pm 3.4$ | $10.8 \pm 3.3$ | $35.8 \pm 2.3$ | $34.2 \pm 2.3$ |
| NeuralOp (FNO-s) | $39.5 \pm 3.2$ | $43.4 \pm 5.8$ | $42.8 \pm 2.0$ | $13.9 \pm 4.1$ | $49.1 \pm 2.0$ | $45.1 \pm 2.6$ |
| NeuralOp (FNO-b) | $41.5 \pm 2.8$ | $49.0 \pm 4.4$ | $43.6 \pm 1.9$ | $14.3 \pm 4.7$ | $50.2 \pm 1.9$ | $46.7 \pm 2.6$ |
| NeuralOp (UNO-s) | $39.2 \pm 3.2$ | $42.0 \pm 6.8$ | $42.6 \pm 2.0$ | $13.1 \pm 4.4$ | $48.4 \pm 2.0$ | $44.8 \pm 2.6$ |
| NeuralOp (UNO-b) | $40.1 \pm 3.1$ | $45.5 \pm 5.3$ | $43.2 \pm 1.9$ | $13.7 \pm 4.6$ | $49.2 \pm 1.9$ | $45.7 \pm 2.4$ |
| | | | **SSIM ↑ (%)** | | | |
| Mamba (SU-s) | $93.6 \pm 1.4$ | $98.4 \pm 1.0$ | $81.4 \pm 9.3$ | $40.8 \pm 18.3$ | $99.5 \pm 0.3$ | $91.1 \pm 2.7$ |
| Mamba (SU-b) | $93.5 \pm 1.7$ | $98.6 \pm 0.8$ | $82.3 \pm 9.1$ | $41.4 \pm 17.6$ | $99.5 \pm 0.3$ | $93.9 \pm 3.5$ |
| Mamba (Seg-s) | $91.4 \pm 2.8$ | $97.6 \pm 1.9$ | $81.4 \pm 8.7$ | $32.8 \pm 17.4$ | $99.5 \pm 0.6$ | $92.4 \pm 4.4$ |
| Mamba (Seg-b) | $90.1 \pm 3.1$ | $97.6 \pm 1.6$ | $75.2 \pm 11.1$ | $32.5 \pm 17.1$ | $99.5 \pm 0.6$ | $93.1 \pm 3.6$ |
| Transformer (SU-s) | $89.4 \pm 3.6$ | $96.6 \pm 2.6$ | $81.3 \pm 9.0$ | $34.0 \pm 18.9$ | $99.4 \pm 0.7$ | $92.1 \pm 4.7$ |
| Transformer (SU-b) | $92.4 \pm 2.6$ | $98.1 \pm 1.5$ | $81.8 \pm 9.4$ | $36.9 \pm 19.5$ | $99.4 \pm 0.6$ | $92.1 \pm 4.8$ |
| Transformer (FF-s) | $94.1 \pm 1.3$ | $98.4 \pm 0.9$ | $82.1 \pm 9.7$ | $41.7 \pm 18.5$ | $99.5 \pm 0.4$ | $92.1 \pm 5.0$ |
| Transformer (FF-b) | $94.9 \pm 1.0$ | $98.7 \pm 0.8$ | $84.4 \pm 8.9$ | $41.8 \pm 18.6$ | $99.5 \pm 0.4$ | $90.7 \pm 6.4$ |
| Diffusion (DDIM-s) | $68.8 \pm 11.9$ | $78.4 \pm 5.6$ | $81.0 \pm 7.3$ | $30.9 \pm 15.4$ | $6.4 \pm 7.5$ | $77.4 \pm 10.1$ |
| Diffusion (DDIM-b) | $82.2 \pm 6.6$ | $89.9 \pm 10.0$ | $86.7 \pm 7.1$ | $29.0 \pm 16.2$ | $95.5 \pm 4.6$ | $82.4 \pm 6.4$ |
| Diffusion (DDPM-s) | $77.8 \pm 8.4$ | $60.8 \pm 6.8$ | $86.5 \pm 7.2$ | $26.2 \pm 14.5$ | $89.6 \pm 7.1$ | $82.5 \pm 7.1$ |
| Diffusion (DDPM-b) | $79.2 \pm 6.6$ | $67.4 \pm 8.1$ | $84.0 \pm 5.8$ | $27.4 \pm 14.2$ | $85.3 \pm 4.2$ | $65.5 \pm 8.0$ |
| NeuralOp (FNO-s) | $86.8 \pm 4.4$ | $93.9 \pm 4.1$ | $81.3 \pm 7.8$ | $35.5 \pm 15.6$ | $99.3 \pm 0.6$ | $91.5 \pm 4.7$ |
| NeuralOp (FNO-b) | $91.4 \pm 3.3$ | $98.1 \pm 1.9$ | $81.8 \pm 8.7$ | $40.9 \pm 18.6$ | $99.4 \pm 0.5$ | $92.2 \pm 4.9$ |
| NeuralOp (UNO-s) | $86.5 \pm 4.9$ | $92.3 \pm 5.8$ | $80.2 \pm 8.3$ | $31.4 \pm 13.8$ | $99.2 \pm 0.8$ | $91.8 \pm 4.3$ |
| NeuralOp (UNO-b) | $88.1 \pm 3.9$ | $95.5 \pm 3.2$ | $81.3 \pm 8.5$ | $35.4 \pm 16.3$ | $99.3 \pm 0.6$ | $92.0 \pm 4.5$ |

*Table 11*. Robustness benchmarks on combined bone cells under Gaussian **noise** perturbation ($\sigma = 0.05$). The models were trained and evaluated on noisy Mueller matrix inputs to simulate experimental measurement errors. Results are reported as (mean $\pm$ st) for WD-1D per sample, and mean for WD-1D over the test set. Color coding indicates performance rankings: the *top three* models for each modality are highlighted in (1st), (2nd), and (3rd). The worst performance in each column is underlined.

| Models | ImageD | ImageDelta | ImageEta | ImageTheta | ImagePsi | ImageR |
|---|---|---|---|---|---|---|
| | **WD-1d ↓ (per sample)** | | | | | |
| Mamba (SU-s) | $0.59 \pm 0.13$ | $0.13 \pm 0.12$ | $1.45 \pm 0.40$ | $13.6 \pm 9.74$ | $0.37 \pm 0.11$ | $0.73 \pm 0.27$ |
| Mamba (SU-b) | $0.65 \pm 0.14$ | $0.11 \pm 0.15$ | $1.38 \pm 0.42$ | $9.34 \pm 7.31$ | $0.35 \pm 0.12$ | $0.61 \pm 0.37$ |
| Mamba (Seg-s) | $0.79 \pm 0.47$ | $0.31 \pm 0.43$ | $1.46 \pm 0.99$ | $\underline{19.0 \pm 13.8}$ | $0.61 \pm 0.18$ | $0.72 \pm 0.98$ |
| Mamba (Seg-b) | $0.90 \pm 0.47$ | $0.30 \pm 0.36$ | $\underline{1.89 \pm 0.84}$ | $18.8 \pm 13.3$ | $0.72 \pm 0.16$ | $0.63 \pm 0.84$ |
| Transformer (SU-s) | $1.04 \pm 0.48$ | $0.27 \pm 0.25$ | $1.43 \pm 0.52$ | $13.5 \pm 8.99$ | $0.54 \pm 0.26$ | $0.68 \pm 0.53$ |
| Transformer (SU-b) | $0.72 \pm 0.26$ | $0.11 \pm 0.13$ | $1.41 \pm 0.45$ | $13.1 \pm 8.58$ | $0.52 \pm 0.19$ | $0.68 \pm 0.42$ |
| Transformer (FF-s) | $0.58 \pm 0.16$ | $0.10 \pm 0.09$ | $1.40 \pm 0.45$ | $10.6 \pm 7.59$ | $0.53 \pm 0.12$ | $0.68 \pm 0.41$ |
| Transformer (FF-b) | $0.50 \pm 0.16$ | $0.06 \pm 0.10$ | $1.27 \pm 0.47$ | $10.8 \pm 6.83$ | $0.51 \pm 0.13$ | $0.70 \pm 0.42$ |
| Diffusion (DDIM-s) | $\underline{2.84 \pm 2.55}$ | $1.82 \pm 1.94$ | $1.83 \pm 2.20$ | $14.7 \pm 7.97$ | $\underline{4.27 \pm 2.38}$ | $2.14 \pm 3.58$ |
| Diffusion (DDIM-b) | $1.55 \pm 3.60$ | $0.97 \pm 3.33$ | $1.68 \pm 3.64$ | $13.9 \pm 7.27$ | $1.77 \pm 1.77$ | $2.41 \pm 2.17$ |
| Diffusion (DDPM-s) | $2.14 \pm 4.39$ | $\underline{3.07 \pm 3.24}$ | $1.74 \pm 4.75$ | $18.2 \pm 9.56$ | $3.02 \pm 2.21$ | $2.24 \pm 4.58$ |
| Diffusion (DDPM-b) | $1.63 \pm 2.47$ | $2.20 \pm 2.35$ | $1.12 \pm 1.86$ | $12.4 \pm 7.23$ | $2.90 \pm 1.57$ | $\underline{2.53 \pm 1.83}$ |
| NeuralOp (FNO-s) | $1.04 \pm 0.50$ | $0.44 \pm 0.40$ | $1.47 \pm 0.58$ | $15.7 \pm 8.74$ | $0.55 \pm 0.23$ | $0.76 \pm 0.53$ |
| NeuralOp (FNO-b) | $0.81 \pm 0.34$ | $0.16 \pm 0.20$ | $1.43 \pm 0.37$ | $11.2 \pm 7.26$ | $0.54 \pm 0.21$ | $0.68 \pm 0.40$ |
| NeuralOp (UNO-s) | $1.36 \pm 0.57$ | $0.66 \pm 0.62$ | $1.49 \pm 0.64$ | $16.7 \pm 9.27$ | $0.58 \pm 0.30$ | $0.77 \pm 0.64$ |
| NeuralOp (UNO-b) | $1.07 \pm 0.49$ | $0.29 \pm 0.27$ | $1.42 \pm 0.44$ | $13.89 \pm 8.46$ | $0.58 \pm 0.20$ | $0.68 \pm 0.48$ |
| | **WD-1d ↓ (whole test set)** | | | | | |
| Mamba (SU-s) | $0.46$ | $0.18$ | $1.45$ | $11.5$ | $0.36$ | $0.83$ |
| Mamba (SU-b) | $0.56$ | $0.06$ | $1.38$ | $3.20$ | $0.35$ | $0.63$ |
| Mamba (Seg-s) | $0.77$ | $0.31$ | $1.45$ | $17.8$ | $0.61$ | $0.65$ |
| Mamba (Seg-b) | $0.77$ | $0.29$ | $\underline{1.88}$ | $\underline{18.0}$ | $0.72$ | $0.73$ |
| Transformer (SU-s) | $1.01$ | $0.23$ | $1.43$ | $9.42$ | $0.54$ | $0.63$ |
| Transformer (SU-b) | $0.63$ | $0.17$ | $1.41$ | $9.08$ | $0.52$ | $0.62$ |
| Transformer (FF-s) | $0.50$ | $0.15$ | $1.41$ | $6.30$ | $0.53$ | $0.70$ |
| Transformer (FF-b) | $0.44$ | $0.04$ | $1.27$ | $7.57$ | $0.55$ | $0.71$ |
| Diffusion (DDIM-s) | $\underline{2.74}$ | $1.80$ | $1.45$ | $10.9$ | $\underline{4.27}$ | $1.74$ |
| Diffusion (DDIM-b) | $1.10$ | $0.74$ | $1.21$ | $10.6$ | $1.77$ | $1.62$ |
| Diffusion (DDPM-s) | $1.77$ | $\underline{3.07}$ | $1.50$ | $15.2$ | $3.02$ | $1.69$ |
| Diffusion (DDPM-b) | $1.41$ | $2.20$ | $0.37$ | $5.81$ | $2.90$ | $\underline{2.15}$ |
| NeuralOp (FNO-s) | $1.08$ | $0.47$ | $1.42$ | $13.3$ | $0.52$ | $0.64$ |
| NeuralOp (FNO-b) | $0.68$ | $0.05$ | $1.43$ | $8.32$ | $0.53$ | $0.62$ |
| NeuralOp (UNO-s) | $1.22$ | $0.57$ | $1.43$ | $14.1$ | $0.55$ | $0.65$ |
| NeuralOp (UNO-b) | $0.97$ | $0.30$ | $1.41$ | $10.3$ | $0.58$ | $0.62$ |

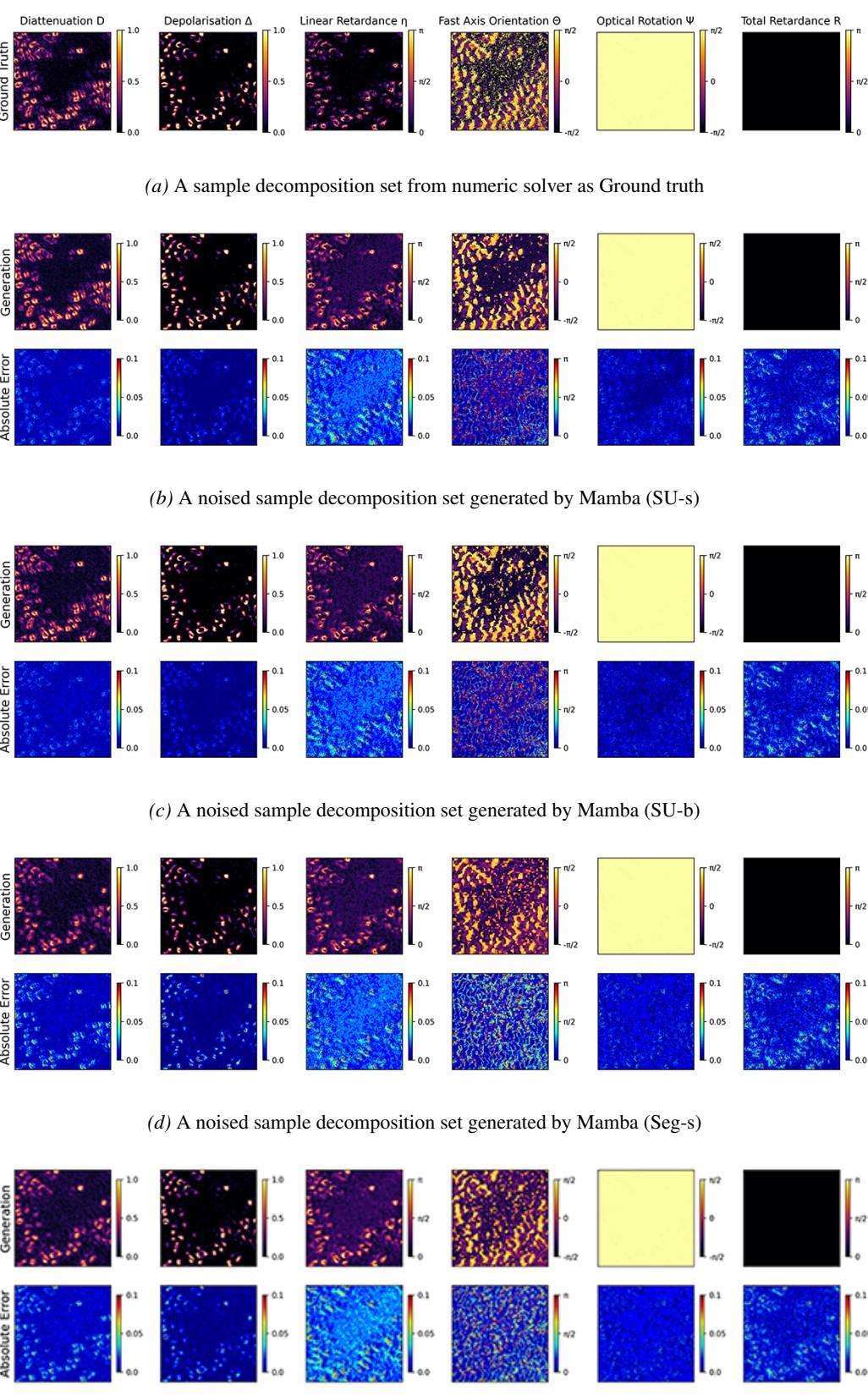

*(a)* A sample decomposition set from numeric solver as Ground truth

*(b)* A noised sample decomposition set generated by Mamba (SU-s)

*(c)* A noised sample decomposition set generated by Mamba (SU-b)

*(d)* A noised sample decomposition set generated by Mamba (Seg-s)

*(e)* A noised sample decomposition set generated by Mamba (Seg-b)

*Figure 9.* A. Qualitative evaluation of noised polarimetric parameter estimation by **Mamba**-based models.

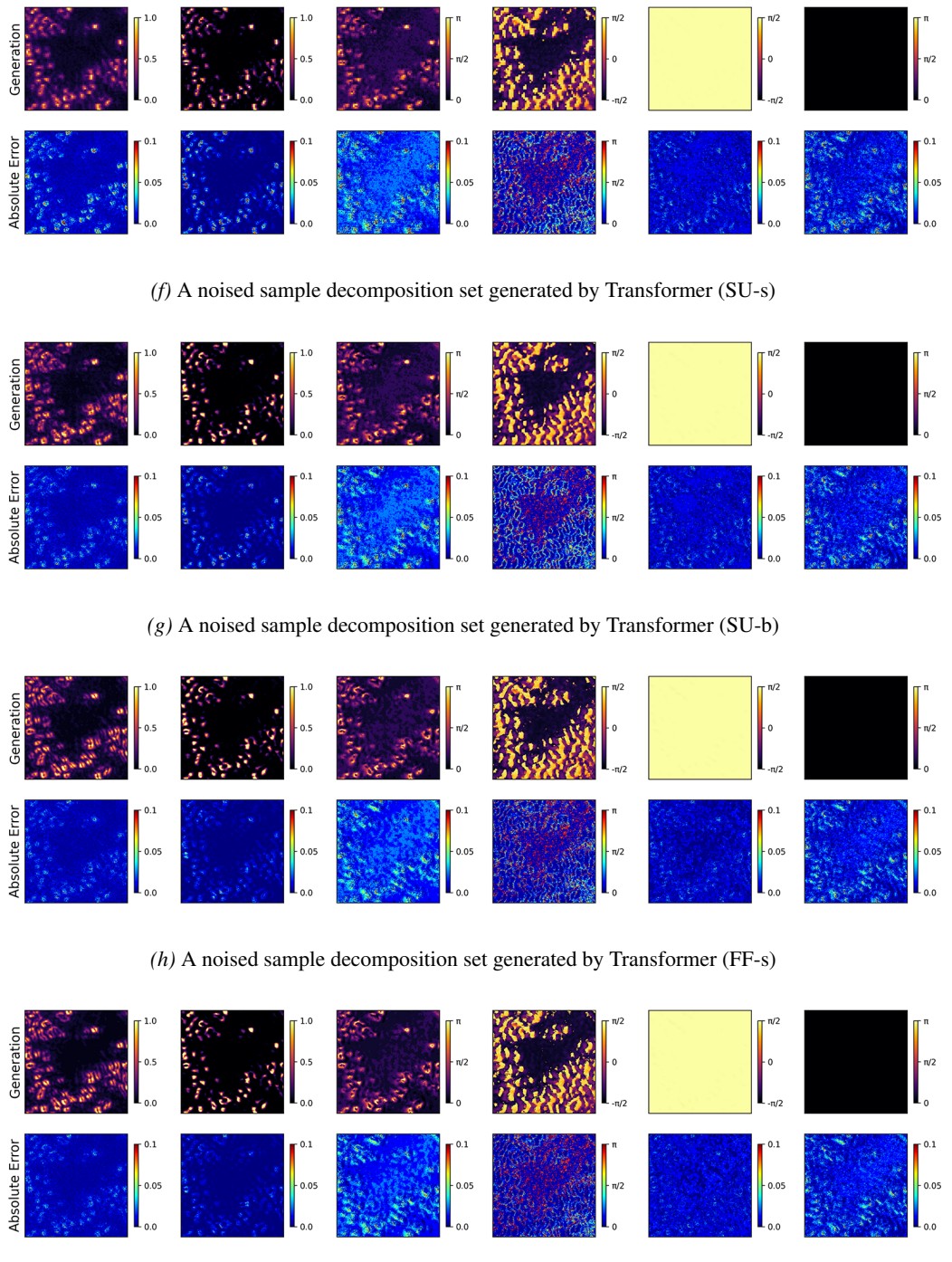

*(f)* A noised sample decomposition set generated by Transformer (SU-s)

*(g)* A noised sample decomposition set generated by Transformer (SU-b)

*(h)* A noised sample decomposition set generated by Transformer (FF-s)

*(i)* A noised sample decomposition set generated by Transformer (FF-b)

*Figure 9.* B. **Qualitative evaluation of noised polarimetric parameter estimation by Transformer-based models.**

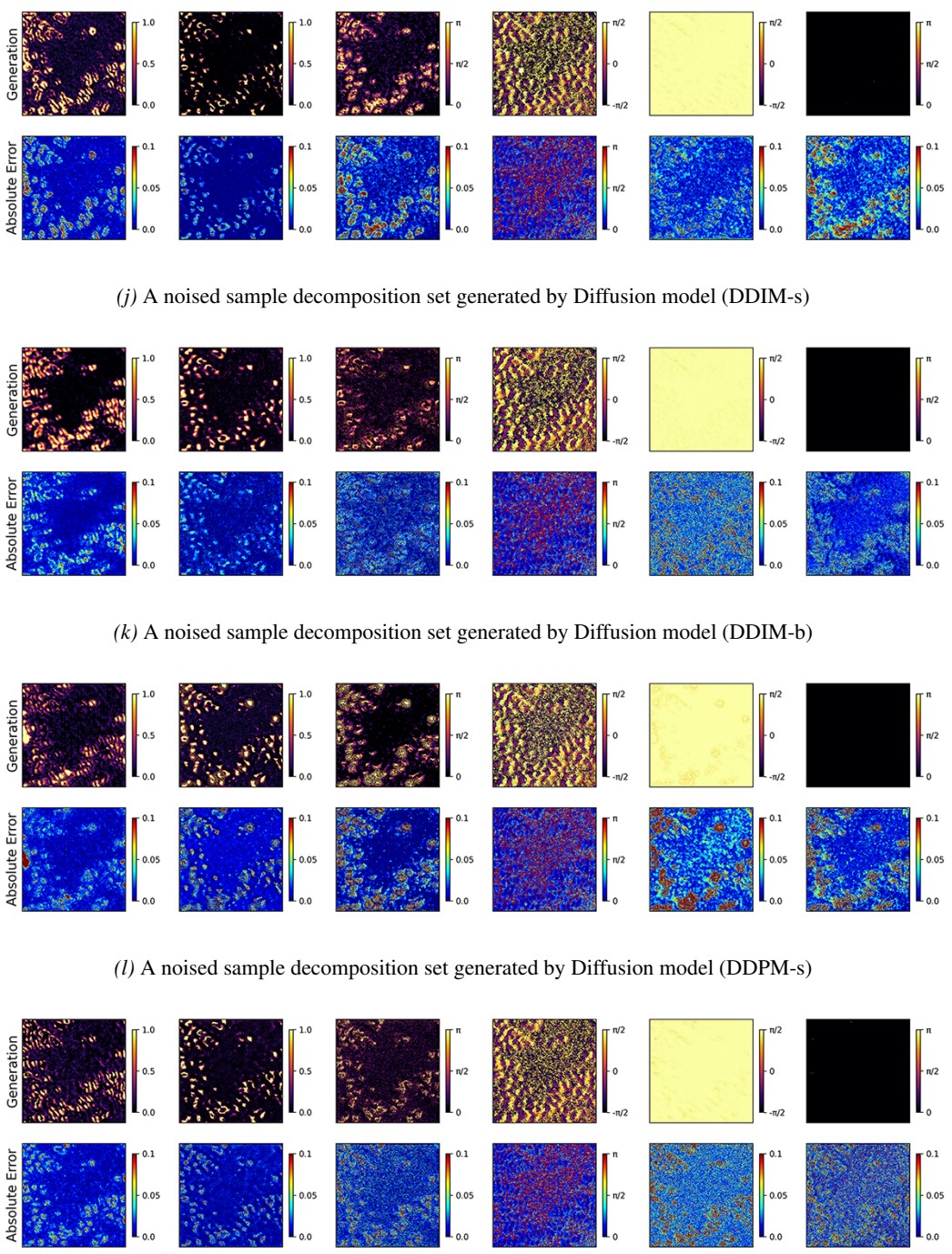

*(j)* A noised sample decomposition set generated by Diffusion model (DDIM-s)

*(k)* A noised sample decomposition set generated by Diffusion model (DDIM-b)

*(l)* A noised sample decomposition set generated by Diffusion model (DDPM-s)

*(m)* A noised sample decomposition set generated by Diffusion model (DDPM-b)

*Figure 9.* C. **Qualitative evaluation of noised polarimetric parameter estimation by Diffusion-based models.**

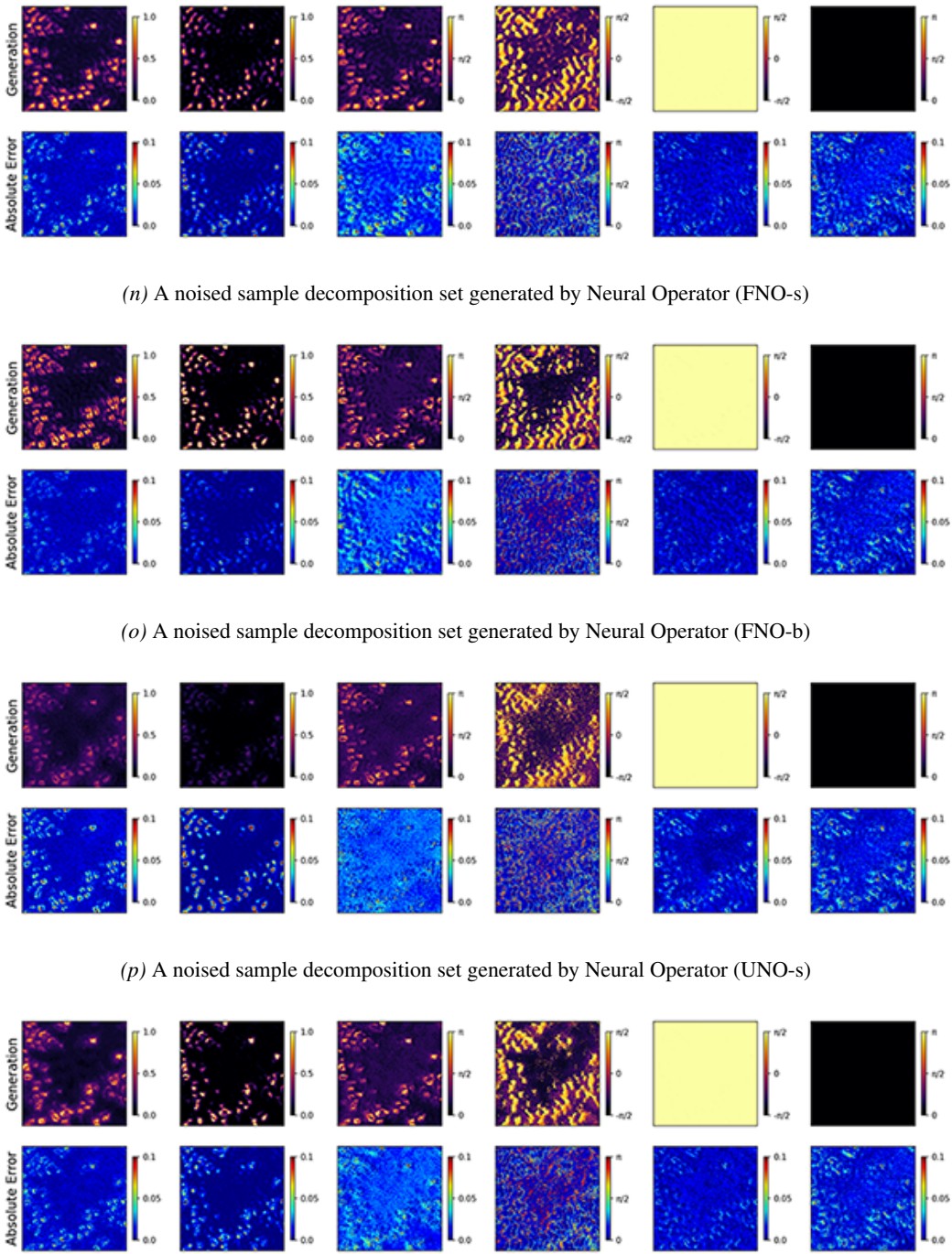

*(n)* A noised sample decomposition set generated by Neural Operator (FNO-s)

*(o)* A noised sample decomposition set generated by Neural Operator (FNO-b)

*(p)* A noised sample decomposition set generated by Neural Operator (UNO-s)

*(q)* A noised sample decomposition set generated by Neural Operator (UNO-b)

*Figure 9.* D. **Qualitative evaluation of noised polarimetric parameter estimation by Neural Operator-based models.** The figure presents a visual comparison across six polarimetric decomposition parameters: Diattenuation (D), Depolarization ($\Delta$), Linear Retardance ($\eta$), Fast Axis Orientation ($\theta$), Optical Rotation ($\psi$), and Total Retardance (R). The rows display the Ground Truth (top), the model's Prediction (1st row of each subfigure), and the corresponding Absolute Error map (2nd row of each subfigure). All predicted values have been remapped to their physical ranges for accurate comparison. The color bars on the right of each map indicate the value scale for that specific parameter and its error.

# H. Source Algorithms

This list includes the official research repositories that MMPD-Bench referenced and adapt.

*Visual Transformers & State Space Models.*

- **Swin-Unet** (Cao et al., 2022): Swin-Unet Transformer, a U-shaped hierarchical vision transformer using shifted-window attention for efficient image modeling. (`https://github.com/HuCaoFighting/Swin-Unet`)

- **FactFormer** (Li et al., 2023b): Factorized Transformer for scalable PDE surrogate modeling via low-rank/factorized attention. (`https://github.com/BaratiLab/FactFormer`)

- **Vim** (Zhu et al., 2024): Vision Mamba backbone leveraging bidirectional state space models (SSMs) for efficient long-sequence visual modeling. (`https://github.com/hustvl/Vim`)

*Generative Models.*

- **DDPM/DDIM** (Ho et al., 2020; Song et al., 2020a): Denoising diffusion models for conditional generation, implemented with classifier-free guidance. (`https://github.com/lucidrains/denoising-diffusion-pytorch`)

*Neural Operators.*

- **FNO** (Li et al., 2020): Fourier Neural Operator for learning solution operators of parametric PDEs via spectral (frequency-domain) representations. (`https://github.com/neuraloperator/neuraloperator`).

- **UNO** (Rahman et al., 2023): U-shaped Neural Operator that combines U-Net-style multiscale feature hierarchies with operator learning for improved multiresolution generalization. (`https://github.com/ashiq24/UNO`)

*Physics Simulation.*

- **Lu–Chipman** (Lu & Chipman, 1996): Standard numerical polar decomposition extracting diattenuation, retardance, and depolarization from Mueller matrices via the approach in Appendix B.

