# OpenReview forum: "MMPD-Bench: Bridging Multimodal Fission with Multi-Polarimetric Modalities Decomposition"
_ICML.cc/2026/Conference — ICML 2026 regular_

### Official Review · Reviewer_qiXm · 2026-02-18

**Soundness:** 3
**Presentation:** 3
**Significance:** 4
**Originality:** 3
**Overall Recommendation:** 4
**Confidence:** 4

**Summary:**

This paper introduces MMPD-Bench, a new benchmark that reframes Mueller Matrix Polar Decomposition (MMPD) as a multimodal “modality fission” problem. The authors release a real-world dataset of 21,412 paired high-resolution Mueller matrix observations and corresponding decomposed polarimetric modalities, and propose a multi-faceted evaluation protocol that covers computational efficiency, reconstruction fidelity, physics-based consistency, and robustness to acquisition noise. They benchmark state-space models, vision transformers, conditional diffusion models, and, for the first time in this domain, neural operators, reporting trade-offs between speed, fidelity, and robustness.

**Compliance With Llm Reviewing Policy:**

Affirmed.

**Final Justification:**

My concerns have been addressed and I kept my score.

**Key Questions For Authors:**

Generalization: How do the findings on bone cell data translate to other materials or non-microscopic Mueller matrix acquisitions?

Physics Integration: Did you consider incorporating the forward physical operator $\mathcal{F}$ into the training loop (e.g., as a physics-informed loss) to enforce inter-channel dependencies?

Denoising Strategy: For the diffusion models, why was Classifier-Free Guidance chosen over other conditioning techniques specifically designed for inverse problems (like DPS or manifold constraints)?

**Limitations:**

The generalization of the datasets.

This paper makes a valuable contribution by introducing a large-scale dataset and a comprehensive framework for multimodal decomposition. However, there are significant issues with the loss functions, evaluation metrics, and dataset splitting that need addressing. While the dataset and methods are promising, the lack of clarity in certain areas (especially around physical constraints and robustness) weakens the paper’s impact

**Strengths And Weaknesses:**

The MMPD-Bench dataset is a valuable community contribution, providing a large-scale, high-resolution collection of Mueller matrix observations. The paper introduces a novel framework for multimodal decomposition and successfully integrates neural operators into polarimetric analysis to some extent. The proposed evaluation protocol offers a comprehensive approach that combines computational efficiency with physical consistency.
---
Strengths

1) Bridging Two Fields: The paper successfully translates a specialized problem in scientific computing (optical polarimetry) into a structured machine learning challenge, making it accessible to the broader AI community.

2) Technical Breadth: The benchmarking is rigorous, covering a wide range of state-of-the-art architectures including Mamba, Swin-Unet, FactFormer, and Fourier Neural Operators (FNO).

3) Insightful Analysis: The paper identifies non-trivial trade-offs, such as how larger model scales do not necessarily imply better robustness and the specific vulnerability of U-shaped architectures to high-frequency noise.

4) Reproducibility: The authors provide an open-source codebase and a large-scale real-world dataset, which is a significant service to the research community.

Major Weaknesses:
1.	Unexplained Loss Functions: One major issue is the lack of explanation regarding the loss functions used, especially in the diffusion models. Based on the results, it seems the authors might be using default losses like L1 or focusing on noise/latent losses in diffusion models, which lack the necessary physical constraints. Since the authors claim that their approach is "physics-guided," this omission raises doubts about the physical foundation of the method.
2.	Inappropriate Metrics for Angular Parameters: Using PSNR/SSIM to evaluate angular parameters such as θ and ψ is clearly inappropriate. These metrics are designed for pixel-wise evaluations, but angular parameters are periodic, requiring circular-aware metrics (like angular error or modulo-based measures). The current approach leads to misleading results, especially for angular fields.
3.	Risk of Data Leakage in Dataset Split: The dataset split method is risky as it uses overlapping patches, which could cause data leakage between the training, validation, and test sets. This could lead to overly optimistic performance results. It's critical that the splits ensure no overlap to avoid this issue.
Minor Weaknesses:
1.	Lack of Forward-Physics Validation: The paper doesn't validate the forward physics — i.e., recomposing the Mueller matrix from the predicted components and measuring re-projection errors. This would have strengthened the physical correctness validation of the model.
2.	Issues with Robustness Evaluation: The robustness evaluation seems limited, as it trains and tests on noisy data. A more standard approach would be to train on clean data and test under unseen noise or perturbations, ensuring a better understanding of how the model performs in real-world conditions.
3.	Unclear Definition of "Retardance Consistency": The concept of "retardance consistency" is not well defined, especially the "Truth (-0.15)" value used in the analysis. Without a clear formal definition and justification, it’s difficult to interpret these results.
4.	Memory Usage and OOM Issues: The article doesn’t provide enough details about the memory usage during inference (e.g., selective-scan kernels, activation checkpointing). These details are crucial for comparing the models fairly, and clearer memory profiling would help validate the claims.

---

> ### Author Rebuttal · Authors · 2026-03-30
>
> # Response to Reviewer qiXm
>
> We thank the reviewer for recognising the uniqueness and importance of our contributions. Due to the response length, we include more information in the link and in the above responses. We appreciate the further reading.
>
> [Reference & Data & Link](https://anonymous.4open.science/r/MMPD_bench-F272/)
>
> ## Major W1: Physics-Guided Training Clarification
>
> Thanks for sharing this concern. We respectfully clarify that MMPD-Bench does not employ physics-guided training or loss design. Instead, the physical basis lies in the decomposition targets, derived from a Mueller matrix (MM) decomposition pipeline.
>
> The benchmark adopts standard training objectives (ε-prediction for DDPM/DDIM and L1 losses for other architectures, following literatures and practices in Appendix C and G) to evaluate how well models approximate this mapping without domain-specific modifications, ensuring fair comparison across model families.
>
> We will include a complete training specification table (`loss, optimizer, schedule, hyperparameters`) in the appendix to improve clarity.
>
> ## Major W2: Metrics for Angular Parameters
>
> We thank the reviewer for raising this important concern.
>
> If we understand the reviewer correctly, the concern is that PSNR/SSIM may be inappropriate for angular quantities related periodicity effects (e.g., wrapping leading to inflated errors). We respectfully clarify this does not happen because:
> - [New Table] In our setting, the angular parameters are defined within bounded physical ranges: θ ∈ [−π/2, π/2) and ψ ∈ [−π/2, π/2), which are also used for de-normalising model outputs. Within these ranges, the representation is continuous without wrap-around discontinuities, so such periodicity-induced artefacts do not arise.
>
> - Furthermore, modalities are evaluated as spatial maps to compare whether tissue sample features are correctly learnt in each modality. PSNR and SSIM are therefore used to assess whether models correctly recover spatial structures and patterns. Meanwhile, retardance consistency is also compared and discussed in Figure 3 & Section 4.2.
>
> We include the range table in Link and highlight this in Appendix B.
>
> ## Major W3: Data Leakage and Dataset Split
>
> We thank the reviewer for this concern. We confirm there is **no data leakage** between train/valid/test sets. The “overlap” in Table 1 refers to the acquisition stage, where the imaging window is translated with 1-directional overlap. In practice, ~2 captures per tissue are selected, and only consecutive captures partially overlap; non-consecutive captures do not. Overlapping regions are removed before patch extraction and do not contribute to the dataset. This also explains why the patch count exceeds the simple (1 − overlap_ratio) estimate. Patching uses a sliding window (stride 200/250) within each sample to maximise coverage.
>
> Hence, the dataset statistics remain unchanged as shown in Link after double-check. We will revise the manuscript to clarify this and avoid ambiguity.
>
> ## Minor W1 & Q2: Forward-Physics Validation & Integration
> Thanks for this insightful suggestion. We agree that forward-physics validation is an important direction, but not practically feasible in MMPD study as shared with a group of inverse problems.
>
> The projection error from the `recomposed MM` is unreliable in practice. First, errors in individual modalities propagate nonlinearly through the non-commutative decomposition ($M = M_Δ · M_R · M_D$) [3,4], so a small re-projection error does not guarantee accurate recovery. Second, the measured MM usually contains noise from detector, calibration, and acquisition processes, and are therefore imperfect references for forward validation [5]. We will clarify this in the manuscript.
>
> Consequently physics integration of forward operator in training is not feasible. Nonetheless, forward-physics integration via data-driven surrogates is a promising direction [6,7]. *Please check the full brainstorming in Link*.
>
> ## Minor W2 & W4: We refer to Reviewer jKgn Q2 & vey1 W3 for details.
>
> ## Minor W3: We clarified the definition and revised Figure 3 in the manuscript.
>
> ## Q1: Generalisation to Other Materials & Acquisition Settings
>
> We thank the reviewer for this question. This is addressed in Reviewer jKgn Q1 and Reviewer y7vF W1&Q1, where we show that the learned mapping generalises across wavelength, sample type, and measurement conditions. Please also refer to our new results in Link.
>
> ## Q3: Denoising Strategy & Conditioning Choice
>
> Thanks for this great question. DPS and manifold constraints rely on an explicit forward operator to enforce measurement consistency. As discussed in Minor W1 & Q2, such a forward model is not practically feasible. Conditional generation with CFG learns the conditional distribution $p(modalities | M)$ directly from paired data without an explicit forward operator, and is tunable upon condition. This enables consistent comparison across model families under a unified setting.

---

> > ### Author Rebuttal · Reviewer_qiXm · 2026-04-05
> >
> > My concerns have been addressed. Some of them are aligned with other reviewers.

---

> > > ### Author Response · Authors · 2026-04-07
> > >
> > > Thanks for the active engagement and the feedback that our rebuttal has fully addressed your concerns. We sincerely appreciate the insightful concerns and follow-up concerns in the review, and the follow-up work in reading the responses and checking the new data in Link.
> > >
> > > Accordingly, many efforts are involved in physically collecting the new data and conducting the computational tests. We hope our answers and efforts help build up more recognition and confidence in contributions and the generalisation of MMPD-Bench.
> > >
> > > Thank you again for your time and support.

---

### Official Review · Reviewer_y7vF · 2026-03-10

**Soundness:** 3
**Presentation:** 3
**Significance:** 3
**Originality:** 4
**Overall Recommendation:** 4
**Confidence:** 3

**Summary:**

This paper introduces MMPD-Bench, a benchmark for multi-polarimetric modalities decomposition, reframing Mueller matrix polar decomposition as a multimodal modality-fission task within modern representation learning. The benchmark provides a dataset of 21,412 high-resolution Mueller matrix observations, standardized evaluation protocols, and comparisons across several model families, including state space models, vision transformers, diffusion models, and neural operators.
Overall, this submission's primary finding pertains to the trade-offs between accuracy, robustness, and computational efficiency across different surrogate architectures. The results show that deterministic models and neural operators achieve strong fidelity and efficiency, whereas diffusion models demonstrate better robustness under noise but struggle to preserve strict physical consistency.

**Compliance With Llm Reviewing Policy:**

Affirmed.

**Key Questions For Authors:**

-	Generalization across imaging systems. How well would models trained on the current dataset generalize to different polarimetric acquisition setups or wavelengths, given that the benchmark data is collected using a single system configuration?
-	Model scaling behavior. The benchmark evaluates small and base variants of several architectures. Could the authors provide further analysis on how performance scales with model size or training data, particularly for architectures such as diffusion models or transformers?

**Limitations:**

Yes.

**Strengths And Weaknesses:**

Strengths

-	Novel problem formulation bridging physics and multimodal learning. The paper reframes Mueller matrix polar decomposition as a modality-fission problem, offering an interesting conceptual bridge between computational optics and multimodal representation learning.
-	First comprehensive benchmark for this task. The work introduces a large real-world dataset (21k samples) and a standardized evaluation protocol covering fidelity, physical consistency, robustness, and computational efficiency, filling an important gap in polarimetric imaging research.
-	Extensive empirical comparison across architectures. The benchmark evaluates four model families (SSMs, ViTs, diffusion models, and neural operators) and provides detailed analysis of the resulting performance trade-offs, offering valuable insights into model behavior for physics-driven inverse problems.

Weaknesses

-	Benchmark focuses on a single application domain. The dataset is derived from bone cell polarimetric imaging, raising questions about whether the conclusions generalize to other polarimetric imaging scenarios.
-	Lack of physics-informed model baselines. Although multiple architectures are evaluated, the benchmark does not include physics-guided neural networks or hybrid physics-ML approaches, which are commonly used in inverse imaging problems.
-	Limited discussion on real-world deployment scenarios. While the paper analyzes runtime and robustness, it provides limited discussion of practical deployment considerations, such as real-time constraints or hardware integration in polarimetric imaging systems.

---

> ### Author Rebuttal · Authors · 2026-03-30
>
> # Response to Reviewer y7vF
>
> We thank the reviewer for recognizing the uniqueness and importance of the problem, acknowledging the dataset contribution and the comprehensive evaluation. We hope this rebuttal communicates the work and plan better.
>
> [Validations & Link](https://anonymous.4open.science/r/MMPD_bench-F272/)
>
> ## W1 & Q1: Single application domain and cross-system generalization
>
> Thanks for raising these points. Generalisation is also addressed in Reviewer jKgn Q1. The mapping from Mueller matrices to decomposed modalities is governed by shared physical principles across samples, materials, and acquisition setups, and the benchmark evaluates approximation of this transformation rather than dataset-specific semantics.
>
> To further assess generalisation, we provide two [additional validations] in Link. First, we evaluate pretrained models (trained at 617 nm) on independent test samples measured at 610 nm, 650 nm, and 690 nm (Tissue exhibits stable and representative polarimetric responses within the 600–700 nm range). Deterministic models maintain strong fidelity (average PSNR 23–31 dB), while diffusion-based models show degraded performance, preserving the model-family ranking observed in-distribution. We also observe that angular modalities (θ, ψ) generalise more robustly than amplitude-related modalities (D, Δ), consistent with their physical dependencies: angular components are primarily geometry-driven, whereas amplitude components are more sensitive to wavelength-dependent optical coefficients.
>
> Second, if the reviewer share additional interest for cross-medium generalisation, please also check response to jKgn Q1, models are evaluated on standard optical components (half- and quarter-wave plates, WPH10M-633 and WPQ10M-633, Thorlabs). Results remain consistent, with high-fidelity models generalising well and diffusion-based models underperforming..
>
> These results together support that the key findings of MMPD-Bench remain qualitatively consistent under distribution shifts in wavelength, sample type, and measurement conditions. We will incorporate these results and clarify the generalisation discussion in the revised manuscript.
>
> ## W2: Lack of physics-guided model baselines
>
> We thank the reviewer for this valuable suggestion. This point is also addressed in Reviewer jKgn W3. Physics-guided approaches require problem-specific designs (e.g., forward model approximation, constraints, specialised objectives) hard to balance in each model family for fair comparisons. MMPD-Bench instead provides a unified benchmark with representative architectures, and our findings motivate such physics-guided approaches as a natural next step. We will clarify this positioning in the revised manuscript.
>
> ## W3: Limited real-world deployment discussion
>
> We thank the reviewer for this suggestion. Appendix A provides close-related details of the physical system hardware, acquisition pipeline, and calibration procedures of the polarimetric imaging platform used in this work.
>
> The primary contribution of MMPD-Bench is to provide a standardised evaluation framework, including curated data, reference decompositions, baseline implementations, and unified metrics, for systematically assessing how well learning-based models approximate MMPD. Our efficiency evaluation directly addresses deployment-relevant considerations in Figure 2,6,7 and Table 4, where we report inference latency scaling across batch sizes, FLOPs, and peak memory, providing practical guidance for architecture selection under throughput and latency constraints.
>
> Full system-level deployment, including hardware integration and end-to-end optimisation of ML-augmented imaging platforms, is beyond the scope of this benchmark. To benefit broader ML and science communities, MMPD-Bench provides a controlled and reproducible foundation for evaluating model behaviour under realistic data and physics principles, and exceptionally welcomes subsequent system-level developments. We will highlight this in the revised manuscript.
>
> ## Q2: Model scaling behavior
>
> We thank the reviewer for this insightful question. We agree that a more systematic analysis of scaling behaviour across model sizes and training data would further strengthen the benchmark. While a scaling study has been addressed in the current submission, our existing results already provide initial insights. As discussed in Section 4.2 (line 410), larger models do not necessarily improve robustness; smaller (-s) variants exhibit lower degradation rates under noise, indicating that scaling does not monotonically lead to better generalisation in this task.
>
> Extending the benchmark to include more fine-grained sweeps over model sizes and training data regimes is surely in progress. Given the substantial computational cost, we plan to include a comprehensive scaling analysis as part of the benchmark release to ensure consistency and reproducibility. We will clarify this planned extensions in the revised manuscript.

---

> > ### Author Rebuttal · Reviewer_y7vF · 2026-04-03
> >
> > The rebuttal has addressed my concerns, so I maintain my initial rating (weak accept).

---

> > > ### Author Response · Authors · 2026-04-03
> > >
> > > We sincerely thank the reviewers for raising the insightful concerns in the review, and the follow-up work in reading the responses and checking the new data in Link. Thanks for the active engagement and the feedback that our rebuttal has fully addressed your concerns.
> > >
> > > Many efforts are involved in physically collecting the new cross-system data and conducting the computational tests. We hope in this discussion period, our answers and efforts help you build up more recognition and confidence in contributions and generalisation of MMPD-Bench.
> > >
> > > We would also be grateful if you would consider updating your confidence accordingly. Thank you again for your time and support.

---

### Official Review · Reviewer_vey1 · 2026-03-12

**Soundness:** 3
**Presentation:** 3
**Significance:** 3
**Originality:** 2
**Overall Recommendation:** 4
**Confidence:** 3

**Summary:**

This paper presents MMPD-Bench, a benchmark for learning-based Mueller Matrix Polar Decomposition from polarimetric observations. The main idea is to cast decomposition as a multimodal prediction problem where a single Mueller matrix input is mapped to multiple output modalities. The benchmark includes a real-world dataset, a set of representative backbone models being benchmarked, and an evaluation metric set including image fidelity, physical consistency, robustness under noise, and inference efficiency. This benchmark paper provides a common testbed for those interested in the task, and an informative comparison of modern vision architectures for physical parameter inference with multi-polarimetric measurements as input.

**Compliance With Llm Reviewing Policy:**

Affirmed.

**Key Questions For Authors:**

What about future work and next steps? The paper should include some discussion of what would come next.

**Limitations:**

Yes

**Strengths And Weaknesses:**

Strengths:

- The paper is technically sound overall. The benchmark dataset construction and empirical comparisons all appear reasonable, and I did not notice major issues with the experimental setup.

- The benchmark is practically valuable, since it provides a common testbed for learning-based Mueller matrix decomposition and compares a broad range of mainstream model backbones.

- The work seems reproducible and usable in practice: the repository with code ready appears in good shape, which increases the likelihood that others can adopt and build on the benchmark.

Weaknesses:

- The paper does not motivate the task strongly enough for a broader audience. It would benefit from a clearer explanation of why this problem matters and what downstream applications or scientific impact improved inference could enable.

- The significance is somewhat under-communicated. While the benchmark itself is useful, the paper could do more to highlight how progress on this task may matter beyond this specific subcommunity.

- It is unclear whether pretrained weights will be released in addition to code. Training details like number of epochs, choice of optimizers, and compute requirement are also not clear in the paper itself.

---

> ### Author Rebuttal · Authors · 2026-03-30
>
> # Response to Reviewer vey1
>
> We thank the reviewer for recognising that the uniqueness and importance of the problem, acknowledging the dataset contribution and the comprehensive evaluation. We hope this rebuttal communicates the work and plan better.
>
> [Data & Link](https://anonymous.4open.science/r/MMPD_bench-F272/)
>
> ## W1&W2: Task Motivation and Broader Impact
>
> We thank the reviewer for these points. MMPD-Bench targets a broader AI4Science audience, including vectorial optics, medical imaging, material science, and multimodal learning.
>
> MMPD is a fundamental step in polarimetric imaging, extracting physically meaningful parameters that characterise microstructure and intrinsic optical properties, which are difficult to measure directly by other means. However, its computational cost limits real-time and large-scale deployment. Efficient and physically faithful surrogates would enable scalable polarimetric imaging, impacting applications in biomedical imaging and material characterisation. This benefits practical deployment in scenarios such as large-area imaging and high-throughput medical inspections, where the potential of polarimetry are significantly undermined. Beyond this domain, the modality fission formulation generalises to a class of inverse problems where a single observation is decomposed into multiple coupled modalities, as a structural challenge encountered in video understanding, cancer diagnosis, and non destructive testing (NDT).
>
> MMPD-Bench also provides multifaceted values to the ML community. It enriches the modality fission branch of multimodal learning, and offers architectural insights across 4 major methods in preserving inter-modality consistency and the advantage of spectral parameterisation for angular quantities. Unlike existing multimodal benchmarks with semantic relationships, MMPD-Bench is governed by explicit physical laws, enabling study of important topics in multimodal representaion, such as inter-modality consistency, hallucination, and interpretability. This provides a controlled testbed under physically governed principles, making it distinct from existing multimodal benchmarks. Such contributions allow quantitative evaluation of inter-modality relationships under known physical criteria.
>
> We will strengthen this cross-disciplinary positioning in the manuscript.
>
> ## W3: Reproducibility and Training Details
>
> We thank the reviewer for this suggestion. We will release pretrained weights for all benchmarked models via the Hugging Face Hub, alongside the open-source codebase on GitHub. We will also include a comprehensive table of training details in the appendix, including training iteration, optimizer configurations, learning rate schedules, batch sizes, and compute requirements. These additions assure full reproducibility of the benchmark.
>
> ## Q1: Future work and next steps
>
> We thank the reviewer for raising this point. We view MMPD-Bench as a standardised testbed that helps structure future research on multimodal fission. Our findings suggest three future directions:
>
> (1) **Physics-aware training objectives.** Standard loss functions (e.g., MSE, L2) do not adequately capture inter-modality consistency. Future work can incorporate polarimetric constraints to guide physically meaningful objective design, and involve information-theoretic analysis upon cross-modality alignment besides error metrics.
>
> (2) **Data scaling for generalisation.** Extending the benchmark to diverse materials, tissue types, and acquisition configurations, together with calibrated physical phantoms with known polarimetric properties and additional annotations, would enable more comprehensive learning and evaluation of transferability, robustness, and domain adaptation in real-world scenarios. *As a first step*, we have collected multi-wavelength bone-cell (610–690 nm, new samples) and standard wave plate data, demonstrating the extensibility of the benchmark (in Link).
>
> (3) **Advanced architectures.** The observed trade-offs across model families provide guidance for designing efficient computing architectures (e.g., few/1-step diffusion models) for high-resolution scientific, and data-efficient architectures under sparse observations.
>
> We will include a dedicated discussion of these directions in the revised manuscript. More broadly, we aim for MMPD-Bench to support a shift from isolated, application-specific studies towards reproducible machine learning research under shared evaluation protocols.
>
> We expect such developments to benefit a range of application domains where polarimetric measurements are critical, including quantitative bioanalysis, material characterisation, and NDT, where efficient and reliable extraction of coupled physical parameters is essential.
>
> We aim for MMPD-Bench to serve as a common evaluation protocol for future research in this direction. We will also provide standardised data splits and evaluation scripts to facilitate direct comparison and adoption in future work.

---

> > ### Author Rebuttal · Reviewer_vey1 · 2026-04-03
> >
> > Rebuttal has addressed my concerns.

---

> > > ### Author Response · Authors · 2026-04-03
> > >
> > > We sincerely appreciate your efforts in reading and checking the new data. Thanks for the active engagement and the feedback that our rebuttal has fully addressed the concerns.
> > >
> > > We hope in this discussion period, our answers and efforts help you build up more recognition and confidence in MMPD-Bench. We would be very grateful if you would consider updating your rating accordingly.
> > >
> > > Thank you again for your time and support.

---

### Official Review · Reviewer_jKgn · 2026-03-12

**Soundness:** 2
**Presentation:** 3
**Significance:** 3
**Originality:** 3
**Overall Recommendation:** 4
**Confidence:** 2

**Summary:**

This paper introduces a multimodal “fission” benchmark, MMPD-Bench, for predicting multiple polarimetric decomposition modalities from Mueller matrix observations using modern vision architectures, diffusion models, and neural operators. The dataset appears substantial, with 21,412 paired samples from real polarimetric measurements, and the evaluation protocol goes beyond standard image metrics by incorporating physics-inspired consistency checks.

**Compliance With Llm Reviewing Policy:**

Affirmed.

**Final Justification:**

My concerns are addressed and accordingly I increase my rating.

**Key Questions For Authors:**

1, The benchmark is built only from bone-cell data, so it is unclear whether the conclusions generalize to other tissues or optical media.

2, Robustness is tested only with Gaussian noise, which may not reflect the more complex non-Gaussian noise encountered in real polarimetric imaging. How can this method being adapted to other practices.

3, Comparing only vanilla diffusion models may be unfair for this physics-constrained problem, since physics-guided variants could behave differently. Can the authors test on more relevant diffusion baselines?

**Limitations:**

The discussion of limitations and negative societal impact is fairly brief in its current state.

**Strengths And Weaknesses:**

Strengths:

1, The problem is interesting and reasonably important. Reframing MMPD as a standardized ML benchmark seems useful. The paper argues that existing computer vision benchmarks do not capture the Stokes-Mueller structure, and therefore a new benchmark is demanded.

2, The dataset contribution also appears meaningful. The paper uses real wide-field Mueller polarimetric measurements from various bone-cell tissue sections and reports explicit split counts totaling 21,412 paired samples. The acquisition description in the appendix also gives helpful instrumentation details.

3, Comprehensive evaluations on this benchmark from three practical perspective are present in this paper: efficiency, modality fidelity and robustness to acquisition perturbations.

Weakness:

1, My main concern is novelty at the methodological level. The paper is primarily a benchmark paper that adapts existing model families rather than proposing a new method. The “modality fission” framing is interesting, but in its current form it reads more like a conceptual re-labeling of a structured inverse problem to a specific domain.

2, A second concern is that the benchmark uses solver outputs as targets, so the models are learning to approximate a decomposition pipeline rather than recover independently measured physical ground truth. Making some claims about “physical fidelity” seem somewhat overstated.

3, Another weakness is the lack of stronger model-based inverse-problem baselines. Since the paper explicitly formulates MMPD as a physics inverse problem, it would be valuable to compare against optimization-based or hybrid methods, such as deep unfolding or other physics-guided reconstruction approaches, rather than limiting the benchmark mostly to learned surrogate families.

---

> ### Author Rebuttal · Authors · 2026-03-30
>
> # Response to Reviewer jKgn
>
> We thank the reviewer for recognising the importance and contributions of MMPD-Bench. We hope this rebuttal builds a better understanding of the work and plan.
>
> [Reference & Data & Link](https://anonymous.4open.science/r/MMPD_bench-F272/)
>
> ## W1: Methodological Novelty
>
> Thanks for sharing the concern. MMPD is a well-established scientific computing problem. As a benchmark, MMPD-Bench does not introduce new models, but a new angle and reframes MMPD by modality fission (MF) in multimodal learning [1], bridging a unified formulation for joint prediction of multiple coupled physical quantities. This enables systematic evaluation of model behaviour (cross-modality dependencies, output consistency, fidelity and robustness) and connects to standardised evaluation protocols in machine learning. Accordingly, MMPD-Bench selects representative architectures across four model families to establish baselines as foundations to support subsequent development of specialised approaches (e.g., physics-informed and hybrid methods). We will highlight this positioning in the manuscript.
>
> ## W2: Physical Fidelity vs Solver-Derived Targets
>
> Thanks for this insightful point. In polarimetric imaging, decomposed physical modalities are not directly measurable. Detectors measure intensity, and polarimetric information is reconstructed from measurements under varying states. Our Mueller matrix (MM) is obtained via calibrated multi-shot acquisition, and its decomposition follows established inverse formulations [2], widely used to derive physical modalities. These quantities are therefore not directly observable but obtained through nonlinear reconstruction. Accordingly, solver targets correspond to the defined physics quantities, and “physical fidelity” refers to consistency with it. We will clarify this in the manuscript that the evaluation emphasises physical consistency within this framework.
>
> ## W3: Absence of Physics-informed Inverse Problem Baselines
>
> Thanks for this suggestion. Physics-informed approaches (e.g., Deep Unfolding, physics-guided reconstruction) are relevant to inverse problems. But in MMPD, the forward mapping to a reconstructed MM is not feasible directly from inversion and requires external approximation. Such task-specific designs, along with constraints and specialised objectives, fall outside standardised baseline comparisons. MMPD-Bench instead aims to establish a unified benchmark for evaluating model families under a consistent setting. We view physics-informed and hybrid approaches as important follow-up directions based on this benchmark, and will clarify this in the manuscript.
>
> ## Q1: Generalisation on Mediums Beyond Bone-Cell
>
> We appreciate this concern. The benchmark uses bone-cell data (healthy and diseased) as a controlled large-scale real-measurement testbed with complex polarimetric characteristics. The mapping from MMs to decomposed modalities follows the same physical principles across samples and materials. Accordingly, the benchmark evaluates how well models approximate this transformation.
>
> [NEW DATA] To provide cross-medium evidence, we evaluated baselines trained on bone-cell data on standard optical components (wave plates). Results are consistent: models with high fidelity on bone-cell data also perform well on wave plates, while vanilla DDPM/DDIM results remain less reliable. This suggests the findings are not specific to the biological domain.
>
> Results are provided in Link. Covering more diverse tissues, materials, and systems is an important direction for future work, and will be clarified in the manuscript.
>
> ## Q2: Noise Model and Robustness Evaluation
>
> Thanks for this insightful point. Models are trained and evaluated on *real* MM measurements, which include non-Gaussian noise from extra systematic/random errors and even aberrations. Thus, MMPD-Bench reflects realistic conditions beyond purely synthetic noise. The Gaussian noise experiment (Section 4.2) serves as a controlled diagnostic. Other noise types (e.g., Poisson) can be incorporated without changing the task formulation. More structured distortions (e.g., missing data or miscalibration) correspond to different problem settings beyond the current scope. We will clarify this in the manuscript.
>
> ## Q3: Diffusion Baselines and Physics-Guided Variants
>
> Thanks for this suggestion. MMPD-Bench does not impose physics constraints in training or loss design. Physics-guided diffusion variants are relevant but require problem-specific designs outside standardised comparisons (see W3). We therefore use representative diffusion models (DDPM/DDIM) as consistent references alongside other general-purpose architectures under a unified setting. And we clarify physics-guided diffusion is an important future direction.
>
> #### Additional comment on limitations
>
> We acknowledge the need for clearer discussion of limitations and will expand the impact statement. This does not affect the core contributions of MMPD-Bench.

---

> > ### Author Rebuttal · Reviewer_jKgn · 2026-04-03
> >
> > Thank you the authors for addressing my concerns, I have no questions at this point.

---

> > > ### Author Response · Authors · 2026-04-03
> > >
> > > We sincerely appreciate your efforts in raising the insightful concerns in the review, and the followup work in reading the responses and checking the new data in Link. Thanks for the active engagement that our rebuttal has fully addressed the concerns and increasing your rating.
> > >
> > > Many efforts are involved in physically collecting the new optical media data and conducting the computational tests. We hope in this discussion period, our answers and efforts help you build up more confidence in contributions and generalisation of MMPD-Bench.
> > >
> > > We are grateful for the positive feedback.
> > > Thank you again for your time and support.

---

### Decision · Program_Chairs · 2026-04-30

**Decision:**

Accept (regular)

**Comment:**

The paper introduces a new benchmark for predicting multiple polarimetric decomposition modalities from Mueller matrix observations. Its contributions include: (a) a new dataset, (b) a unified formulation of Mueller Matrix Polar Decomposition (MMPD) as a modality fission problem, (c) the application of neural operators to MMPD, providing interesting insights, (d) a comparative evaluation of different model architectures and approaches for this problem, and (e) a new evaluation protocol. These contributions were viewed positively by the reviewers.  During the initial review phase, reviewers raised concerns regarding the lack of physics-informed baselines for inverse problems, the limited motivation and under-emphasized significance of the work, and the extent to which the proposed framework generalizes beyond the benchmark considered. In the rebuttal, the authors addressed these concerns satisfactorily.

Overall, I find that the paper makes meaningful contributions, and the benchmark has the potential to become a valuable resource for the computational optics community. Given the reviewers’ assessments and the authors’ rebuttal, I recommend acceptance.